# Sirtuin5 contributes to colorectal carcinogenesis by enhancing glutaminolysis in a deglutarylation-dependent manner

Yun-Qian Wang[1,2,3], Hao-Lian Wang[1,2,3], Jie Xu[1,2,3], Juan Tan[1,2,3], Lin-Na Fu[1,2,3], Ji-Lin Wang[1,2,3], Tian-Hui Zou[1,2,3], Dan-Feng Sun[1,2,3], Qin-Yan Gao[1,2,3], Ying-Xuan Chen[1,2,3] & Jing-Yuan Fang[1,2,3]

Reversible post-translational modifications represent a mechanism to control tumor metabolism. Here we show that mitochondrial Sirtuin5 (SIRT5), which mediates lysine desuccinylation, deglutarylation, and demalonylation, plays a role in colorectal cancer (CRC) glutamine metabolic rewiring. Metabolic profiling identifies that deletion of SIRT5 causes a marked decrease in $^{13}$C-glutamine incorporation into tricarboxylic-acid (TCA) cycle intermediates and glutamine-derived non-essential amino acids. This reduces the building blocks required for rapid growth. Mechanistically, the direct interaction between SIRT5 and glutamate dehydrogenase 1 (GLUD1) causes deglutarylation and functional activation of GLUD1, a critical regulator of cellular glutaminolysis. Consistently, GLUD1 knockdown diminishes SIRT5-induced proliferation, both in vivo and in vitro. Clinically, overexpression of SIRT5 is significantly correlated with poor prognosis in CRC. Thus, SIRT5 supports the anaplerotic entry of glutamine into the TCA cycle in malignant phenotypes of CRC via activating GLUD1.

[1] Division of Gastroenterology and Hepatology, Renji Hospital, School of Medicine, Shanghai Jiao Tong University, 145 Middle Shandong Road, Shanghai 200001, China. [2] Key Laboratory of Gastroenterology and Hepatology, Ministry of Health, Renji Hospital, School of Medicine Shanghai Jiao Tong University 145 Middle Shandong Road, Shanghai 200001, China. [3] State Key Laboratory for Oncogenes and Related Genes, Shanghai Institute of Digestive Disease, Renji Hospital, School of Medicine Shanghai Jiao Tong University 145 Middle Shandong Road, Shanghai 200001, China. Correspondence and requests for materials should be addressed to Y.-X.C. (email: yingxuanchen71@sjtu.edu.cn)

The aberrant metabolic characteristics of a tumor are regarded as a hallmark of cancer and have emerged as an attractive target for novel therapeutic strategies. As the third most commonly diagnosed malignancy and the fourth leading cause of cancer-related death worldwide[1,2], colorectal cancer (CRC) also shows deregulated metabolic profiling during tumorigenesis[3,4]. However, the precise mechanism remains unclear.

The Warburg effect was the first identified metabolic alteration in tumors, and refers to a high rate of glycolysis to convert the majority of glucose to lactate, irrespective of sufficient oxygen[5]. Besides glucose, glutamine is another essential growth-supporting substrate in diverse types of cancer[6]. It is consumed and used to refill the pool of precursor molecules for lipid, nucleic acid, and amino-acid synthesis in most transformed cells[6–8]. Mechanisms including several oncogenic mutations or alterations have been shown to regulate cancer glutamine metabolism in a tightly controlled fashion. Deregulation of glutamine metabolism was observed in CRC, associated with PIK3CA mutations that render cancer cells more dependent on glutamine by upregulating glutamate-pyruvate transaminase 2 (GPT2)[9]. Oncogene MYC was reported to induce the expression of glutamine transporters and glutaminase (GLS)[10,11]. Activated RAS signaling is also required to drive glutamine reprogramming in pancreatic ductal cell adenocarcinoma and CRC[12–14]. To date, inhibitors that target glutamine metabolism, such as bis-2-(5-phenylacetamido-1,2,4-thiadiazol-2-yl) ethyl sulfide, CB-839, and aminooxyacetic acid (AOA), have been reported to decrease cancer proliferation in vivo[15–17]. However, their clinical applications are limited because of adverse side effects and suboptimal chemical properties[8,18], which prompted us to seek a deeper understanding of cancer glutamine reprogramming to explore alternative treatment strategies.

Recently, emerging evidence indicated that metabolic enzymes could be modulated by a number of post-translational modifications (PTMs), including acetylation, succinylation, malonylation, glutarylation, methylation, propionylation, butyrylation, and crotonylation[19,20]. Sirtuin5 (SIRT5), a member of the sirtuin family, is a global regulator of lysine succinylation[21–23], malonylation[21,24], and glutarylation[25]; however, it shows low or undetectable deacetylation activity. These three novel PTMs are classified as short-chain lysine acylations, which are similar to lysine acetylation, but differ in their hydrophobicity, charge, or hydrocarbon chain length[26]. Desuccinylation of isocitrate dehydrogenase 2 (IDH2) and deglutarylation of glucose-6-phosphate dehydrogenase by SIRT5 lead to reduced levels of cellular reactive oxygen species and consequently protect cells against oxidative damage[27]. Altered activities of SIRT5's targets, such as pyruvate dehydrogenase complex and succinate dehydrogenase, are both implicated in cancer cell metabolic dysregulation[23]. Despite recent efforts to identify various metabolic targets of SIRT5 through large-scale proteomic analysis[21–25], there has been little study of the global metabolic alterations regulated by SIRT5 in malignancy. In addition, the biological functions of SIRT5 in CRC remain largely obscure.

In the present study, we reported that upregulation of SIRT5 in CRC is associated independently with poor outcome of patients with CRC. Via integration of high-throughput gas chromatography mass spectrometry (GC-MS) screening and $^{13}$C-based metabolic flux assay, we identified glutamine-dependent anaplerosis into the tricarboxylic-acid (TCA) cycle as the major metabolic pathway regulated by SIRT5 in CRC cells. Consistently, our in vitro and in vivo results showed that GLUD1, an enzyme involved in glutaminolysis, is critical for SIRT5-driven cancer progression. We also revealed the regulatory effect of SIRT5 on GLUD1 deglutarylation and functional activation. Our findings indicated that SIRT5 is a potential therapeutic target in CRC.

## Results

**SIRT5 is overexpressed in CRC tissues and cell lines.** The SIRT5 protein level in paired normal colon mucosa and cancer tissues was determined by immunofluorescence histochemistry. As shown in Fig. 1a, SIRT5 was strongly positive in CRC compared with the corresponding normal tissues, which were weakly or moderately stained. The difference was significant ($P < 0.0001$, Student's $t$-test) between 84 pairs of CRC (mean intensity score = 29.4; 95% confidence interval (CI) = 27.2–31.6) and normal tissues (mean intensity score = 17.4; 95% CI = 16.4–18.5; Fig. 1b). We identified that the green fluorescence caused by SIRT5 was superimposed with the red fluorescence caused by Mito-track (an anti-Mitochondria antibody), suggesting that the majority of SIRT5 is located in mitochondria in CRC tissues (Fig. 1c). The upregulation of the SIRT5 mRNA level in CRC was also validated independently in two published microarray data sets (GSE 68468 with 262 CRC and 55 normal samples, and GSE 41258 with 186 CRC and 54 normal samples, Fig. 1d and Supplementary Fig. 1a, respectively). A similar trend was observed in 41 paired tumor and adjacent normal tissues in GSE 68468 (Fig. 1e). Furthermore, western blotting showed significantly lower levels of SIRT5 in normal colon epithelial cells (FHC) compared with that in a panel of CRC cell lines (Fig. 1f). The ubiquitous expression of SIRT5 at high levels in cancerous tissues and CRC cell lines suggested that SIRT5 might be involved in CRC progression.

Furthermore, the abundance of SIRT5 (median split) was evaluated in 88 CRC patients with different clinicopathological features. We found that SIRT5 overexpression was associated positively with larger tumor size ($P = 0.036$), increased lymph node metastasis ($P = 0.016$), advanced tumor stage ($P = 0.038$), and American Joint Committee on Cancer (AJCC) stage ($P = 0.005$). However, we observed no significant correlation between SIRT5 and histological grade ($P = 0.322$) and distant metastasis ($P = 0.53$, all comparisons by $\chi^2$-test, Supplementary Table 1). Higher expression of SIRT5 also correlated with shorter overall survival (hazard ratio = 3.45, 95% CI = 1.76–7.10, $P = 0.0004$, Mantel–Cox test, Fig. 1g). As summarized in Fig. 1h, after adjustment for age, gender, histological grade, and AJCC stage, multivariate Cox proportional hazards regression analysis confirmed that SIRT5 upregulation is independently associated with higher risk of mortality in patients with CRC, with an average follow-up of 5 years (hazard ratio = 3.06, 95% CI = 1.37–6.87, $P = 0.007$). We next constructed receiver-operating characteristic (ROC) curves to evaluate the accuracy of survival prediction models based on SIRT5 intensity and AJCC stage. Notably, a combination of SIRT5 and AJCC stage optimized the area under an ROC curve value (0.71 for AJCC-based prediction; 0.77 for combination-based prediction; Fig. 1i). These results indicated that SIRT5 expression correlates with poor prognosis of CRC.

**SIRT5 silencing inhibits CRC cell proliferation.** To assess the role of SIRT5 in the regulation of cancer cell proliferation and survival, SIRT5 expression was knocked down specifically using two short interfering RNAs (siRNAs) in HCT116 and LoVo cells, which showed higher levels of SIRT5 in Fig. 1f. The cell counting kit-8 (CCK-8)-based cell viability test revealed that SIRT5 deficiency led to marked inhibition of proliferation in both cell lines (Fig. 2a,b). To clarify the reason for the reduced proliferation, flow cytometry was conducted to detect the effect of SIRT5 on cell cycle transition and apoptosis. As shown in Fig. 2c, d, phycoerythrin-conjugated Annexin V staining showed that suppression of SIRT5 increased the proportion of apoptotic cells

significantly compared with that in the controls. We further confirmed this by western blotting, in which levels of apoptosis indicators, including cleaved caspase 3, caspase 8 (active fragment p18), caspase 6, PARP, and the DNA damage marker γH2AX, were upregulated after *SIRT5* knockdown (Fig. 2g). In addition,

SIRT5 depletion induced both G2/M phase and S phase arrest of the cell cycle (Fig. 2e, f). In line with the flow cytometry, western blotting analysis showed that depletion of SIRT5 resulted in a dramatic accumulation of cyclin E1 and cyclin A2, which was correlated with the arrest of cell division cycle at the S phase

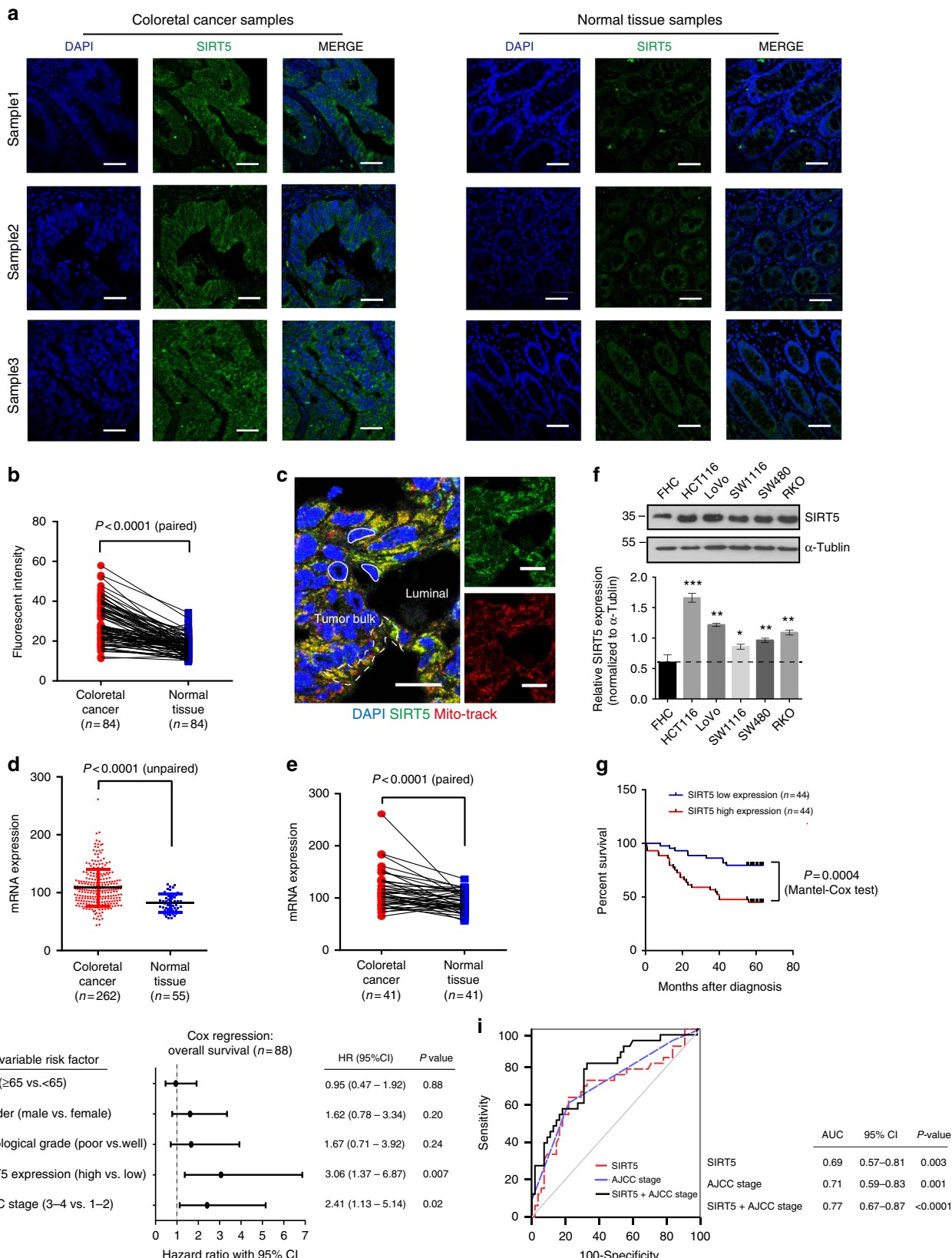

checkpoint. In addition, we detected a reduction in G1 phase regulators, including cyclin D1, cyclin D3, and CDK4, while the expression of cyclin B1 and p21 remained unchanged upon SIRT5 suppression (Fig. 2g). Furthermore, SIRT5 depletion did not alter the mRNA levels of other mitochondrial sirtuins, confirming the specificity of the *SIRT5* siRNAs (Supplementary Fig. 1b). Thus, SIRT5 silencing inhibited CRC cell proliferation by inducing apoptosis and cell cycle arrest.

**The SIRT5-induced growth depends on its catalytic activity.** Given that SIRT5 has robust lysine desuccinylase, demalonylase, and deglutarylase activities, we tested whether the proliferation induced by SIRT5 is mediated by its catalytic activity. We established stable cell lines expressing the vector control, SIRT5 wild type (SIRT5 WT), and the H158Y mutant plasmid (SIRT5 H158Y), a catalytically inactive mutant without lysine deacylation activity (validation of its expression is shown in Fig. 3a, b)[22,24,25]. We found that ectopic expression of SIRT5 significantly promoted the growth of CRC cells, whereas ectopic expression of the catalytic mutant of SIRT5 did not (Fig. 3c, d). Colony formation assays showed similar trends. Compared with the vector and SIRT5 H158Y transfected cells, we observed roughly twofold more colonies in the SIRT5 WT-overexpressing cells (Fig. 3e–g). The inability of the H158Y mutant to promote cell proliferation indicated that the SIRT5-induced cell growth is dependent on its deacylation activity.

**SIRT5 sustains TCA cycle by enhancing glutaminolysis.** Considering that SIRT5's promotion of tumorigenesis is dependent on its catalytic activity, which may regulate cell metabolism, we determined whether SIRT5 has an effect on CRC metabolism reprogramming. *SIRT5* was silenced in HCT116 cells, and their extracted metabolites were subjected to GC-MS analysis. As shown in supplementary Fig. 2a, we observed contrasting distribution patterns between the control and *SIRT5* siRNA-transfected cells when performing partial least squares discriminant analysis (PLS-DA). Metabolite set enrichment analysis of differentially abundant metabolites[28] revealed that SIRT5 downregulation led to profound alterations in the TCA cycle and glutamine metabolism (Supplementary Fig. 2b), including an increased abundance of glutamate and a reduced α-ketoglutarate (α-KG) level, accompanied by a dramatic decrease in almost all TCA cycle intermediates (Fig. 4a and Supplementary Table 2). As illustrated in Fig. 4b, glutamine is metabolized to α-KG, which provides the critical entry point of carbon to fuel the TCA cycle and supports anabolic processes in cancer. These results led us to hypothesize that SIRT5 might have a role in the regulation of cancer glutamine metabolism, which affects the abundance of TCA cycle metabolites.

To determine whether glutamine uptake is affected by SIRT5, we measured the amount of glutamine in the cell culture media

after *SIRT5* silencing. Glutamine consumption remained relatively constant in both HCT116 and LoVo cell lines (Fig. 4c). We then determined if glutamine conversion was altered. Uniform $^{13}C$-labeled glutamine ($[U-^{13}C_5]$ glutamine) was used to monitor the incorporation of glutamine into TCA cycle intermediates after *SIRT5* silencing. As shown in Supplementary Fig. 2c, metabolic and isotopic steady state was achieved at 24 h of incubation with $[U-^{13}C_5]$ glutamine. To exclude metabolic changes caused by cell growth, the LoVo cell line was used in subsequent experiments, because in this cell line, SIRT5 depletion did not suppress cell proliferation at 24 h after *SIRT5* knockdown (Fig. 2b). Our results revealed that suppression of SIRT5 in LoVo cells did not affect $[U-^{13}C_5]$ glutamine incorporation into glutamate (Fig. 4d); however, it significantly reduced the fraction of α-KG(m + 5; Fig. 4e), which suggested that the elevated level of glutamate was not caused by enhanced synthesis, but resulted from its diminished conversion to α-KG. In support of this hypothesis, direct glutamine contribution to the downstream carbon flux after the α-KG node in the TCA cycle was decreased in *SIRT5* knockdown cells. As shown in Fig. 4f, the fraction of succinate (m + 4), fumarate (m + 4), malate (m + 4), citrate (m + 4), and isocitrate (m + 4) were repressed upon SIRT5 silencing. In addition to TCA cycle intermediates, we observed a robustly reduced contribution of glutamine to aspartate and asparagine, which are non-essential amino acids (NEAAs) derived predominantly from glutamine via the TCA cycle (Fig. 4g). In transformed tumor cells, glutamine can be partially oxidized to pyruvate and lactate via flux through the TCA cycle (Fig. 4b)[29]. We further tested the levels of glutamine-derived pyruvate and lactate. As shown in Fig. 4h, although a small percentage of pyruvate and lactate was derived from glutamine (<5%), suppression of SIRT5 led to a slight, but significant, decrease in pyruvate (m + 3) and lactate (m + 3).

Consistent results were observed in SIRT5-overexpressing LoVo cells. SIRT5 overexpression increased the fraction of m + 5 α-KG significantly (Fig. 4i) and promoted a higher rate of incorporation of $[U-^{13}C_5]$ glutamine into m + 4-labeled TCA cycle intermediates, accompanied by glutamine-derived aspartate and asparagine (Fig. 4j). The relative abundance of different mass isotopologues of each metabolite is presented in Supplementary Fig. 3a and 3b. Moreover, the increase in lactate and pyruvate derived from $[U-^{13}C_5]$ glutamine further confirmed that SIRT5 could enhance the TCA cycle flux (Supplementary Fig. 3c). However, the enzymatically deficient mutant did not exhibit all these functions, indicating that the catalytic activity of SIRT5 is required for the enhanced conversion of glutamate to α-KG. There was no difference in cellular glutamine abundance, further confirming that SIRT5 did not affect glutamine uptake (Supplementary Fig. 3d). In addition to the above-mentioned oxidative metabolism of glutamine, cancer cells can metabolize glutamine-derived α-KG reductively to generate citrate via the reversible IDH reaction[30]. To accurately monitor reductive glutamine

**Fig. 1** SIRT5 expression correlates with poor outcome in patients with colorectal cancer. **a** Immunofluorescent staining for SIRT5 (green) in CRC tissues (left) and the corresponding normal tissues (right). Scale bars indicate 50 μm. **b** Plot representing the statistical analysis of SIRT5 protein levels in 84 CRC samples and their paired normal tissues ($P < 0.0001$, Student's *t*-test). **c** CRC tissues were immunostained for SIRT5 (green) and Mito-track (red); yellow in the merged magnified images (left) indicates that SIRT5 was highly abundant in the mitochondria of the CRC cells. The white solid line delineates the cell nucleus. The white dashed line delineates the cancer cell cytoplasm. Scale bars indicate 25 μm. **d**, **e** mRNA expression of *SIRT5* is upregulated in human CRC (GEO data set GSE 68468) (**d**). Results are mean ± SD. *SIRT5* expression in the same data set but compared with paired-adjacent normal tissue using Student's *t*-test (**e**). **f** SIRT5 protein levels were identified in CRC cell lines relative to the normal human colon cell line FHC by western blotting (upper). Quantification of SIRT5 protein levels normalized to α-tublin (lower). *$P < 0.05$, **$P < 0.01$, ***$P < 0.001$. Results are mean ± SD based on two independent experiments. **g** Kaplan–Meier survival analysis of patients with CRC separated into two groups by the median for the SIRT5 signal. The higher signal is shown in red ($n = 44$), and the lower signal is shown in blue ($n = 44$). *P* values were calculated by a log-rank (Mantel–Cox) test. **h** Multivariate survival analysis was performed using Cox's regression model. The hazard ratio (HR) and 95% confident interval (CI) are plotted for each factor. **i** ROC curve analysis showing the ability of AJCC stage (blue dashed line), SIRT5 level (red dashed line), or the combination of two factors (black line) to predict survival in CRC cases. The area under curve (AUC) with 95% CI and the *P* values are shown in the table on the right

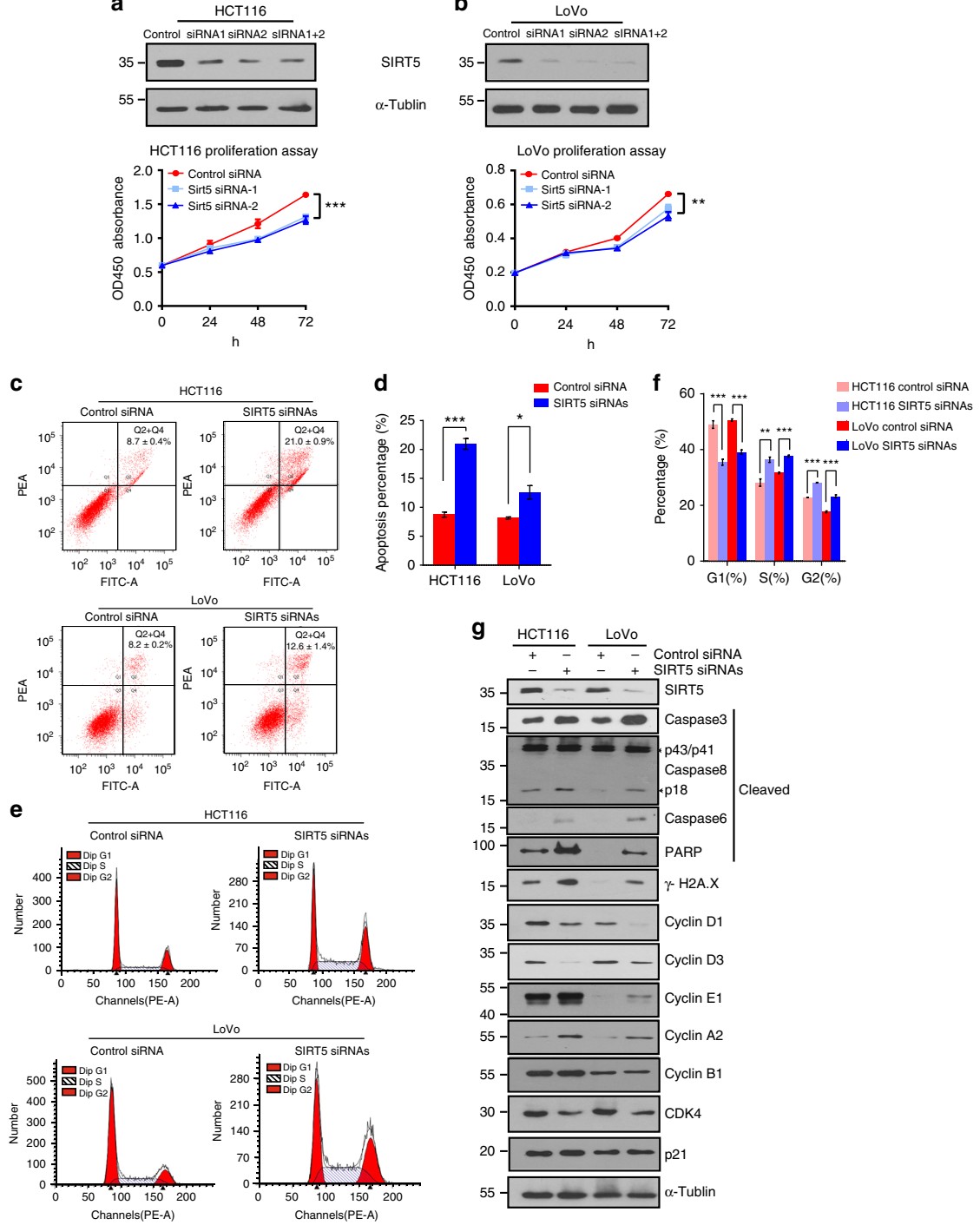

**Fig. 2** SIRT5 is required for proliferation and regulates cell cycle and apoptosis in colorectal cancer cells. **a**, **b** Growth curves of HCT116 (**a**) and LoVo (**b**) cells transfected with two different *SIRT5* short interfering RNAs (siRNAs; blue lines) or control siRNA (red line). Results are presented as mean ± SD of five independent samples. ANOVA with Tukey's test. Western blotting was performed to validate the knockdown efficiency. **c**, **d** Flow cytometric assay based on phycoerythrin-conjugated Annexin V staining showing the increased apoptosis of HCT116 and LoVo cells at 48 h post transfection with *SIRT5* siRNAs. Representative fluorescence-activated cell sorting (FACS) images are shown in **c**. Data in **c** were quantified (**d**). **e**, **f** G2-M phase and S phase arrest were detected in *SIRT5*-knockdown HCT116 and LoVo cells (**e**). Data in **e** were quantified (**f**). Results in **d** and **f** are presented as mean ± SD of three independent samples. Student's *t*-test for **d**. ANOVA with Tukey's test for **f**. *$P < 0.05$, **$P < 0.01$, and ***$P < 0.001$. **g** Western blotting analysis showing increased levels of cleaved caspase 3, caspase 8, caspase 6, PARP, and γH2AX, and altered levels of cell cycle regulators in SIRT5-silenced HCT116 and LoVo cells

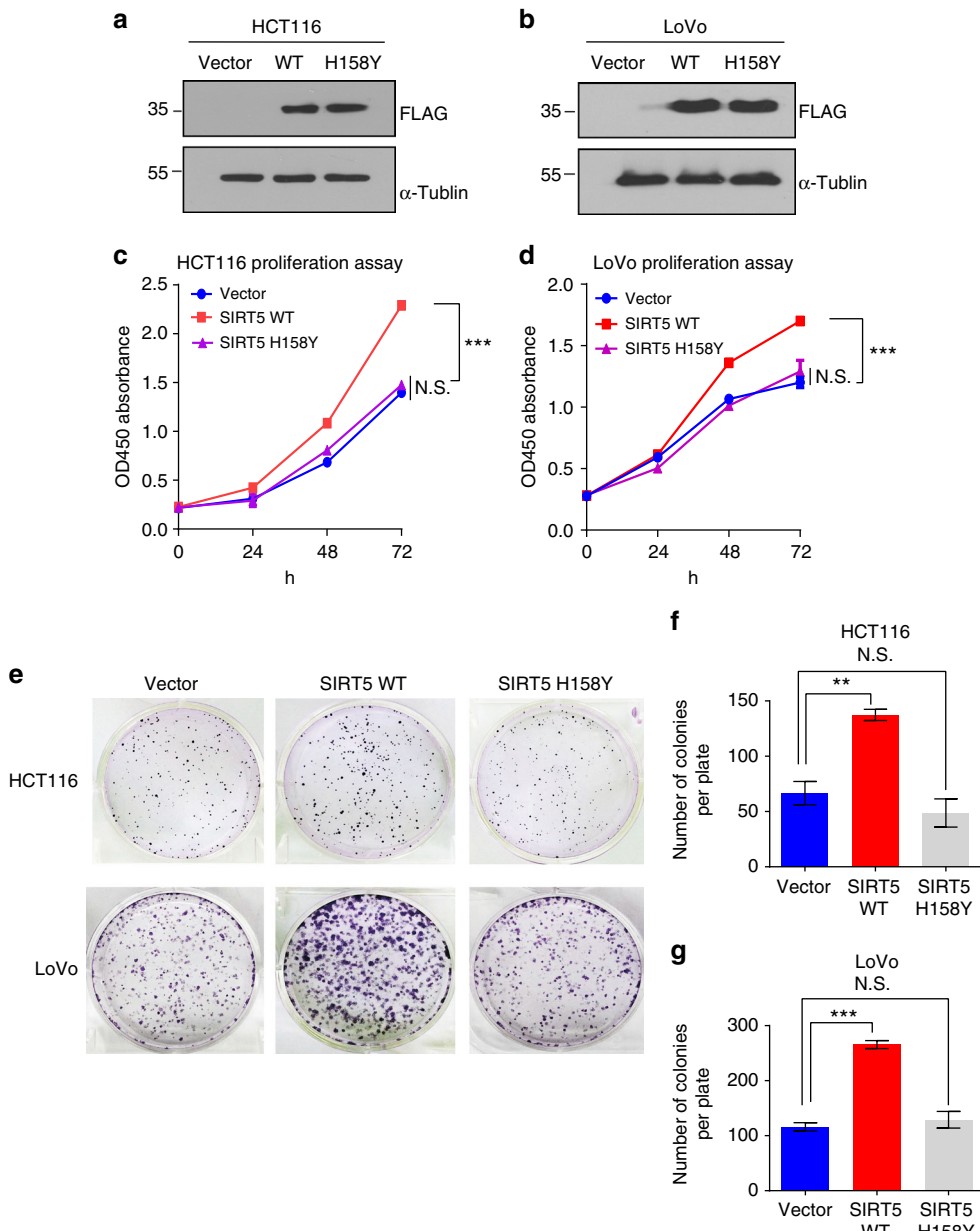

**Fig. 3** SIRT5 promotes colorectal cancer cell growth dependent on its catalytic activity. **a, b** Western blotting confirming the stable expression of the control vector, FLAG-SIRT5 WT, or FLAG-SIRT5 H158Y in HCT116 (**a**) and LoVo (**b**) cells. **c, d** Growth curve of HCT116 (**c**) and LoVo (**d**) cells stably expressing the control vector (blue line), SIRT5 WT (red line), or SIRT5 H158Y (purple line). Data are the mean ± SD of five independent samples. ANOVA with Tukey's test. ***$P < 0.001$. N.S. = not significant for the indicated comparison. **e** Representative wells of the clonogenic growth experiment of HCT116 and LoVo cells stably expressing the control vector, SIRT5 WT, or SIRT5 H158Y, respectively. **f, g** Quantified data of the clonogenic growth experiment in HCT116 (**f**) and LoVo(**g**) cells. Data are the mean ± SD of three independent samples. $P$ values were calculated by ANOVA with Tukey's test. **$P < 0.01$ and ***$P < 0.001$. N.S. = not significant for the indicated comparison

metabolism, we conducted a metabolic flux assay using [1-$^{13}$C] glutamine, which transfers carbon to citrate merely though reductive carboxylation pathway (Supplementary Fig. 4a)[31]. As shown in Supplementary Fig. 4b and 4c, although significant alterations in α-KG (m + 1) were observed, neither knockdown nor overexpression of SIRT5 changed the levels of labeled m + 1 malate, fumarate, and aspartate derived from [1-$^{13}$C] glutamine. Therefore, our results confirmed that SIRT5 mainly regulated the oxidative glutamine metabolism of CRC cells, but had no obvious effect on reductive carboxylation pathway.

Taken together, these results support the view that SIRT5 enhances CRC cells' glutamine utilization in a deacylation-mediated manner, which subsequently increases the refueling of

carbons into the TCA cycle, and might account for SIRT5-induced CRC proliferation.

**SIRT5 promotes glutamine metabolism by activating GLUD1.** Once inside cells, glutamine is converted to glutamate by GLS. Glutamate is then converted to α-KG either via the action of GLUD1 or aminotransaminases (GOT1/2, GPT2, and PSAT1; Fig. 5a). To further identify the mechanism by which SIRT5 promotes the entry of glutamine-derived carbon into the TCA cycle, we studied these enzymes involved in glutamine metabolism. Surprisingly, no significant differences in protein levels of GLS, GLUD1, GOT1/2, GPT2, or PSAT1 were observed upon

SIRT5 knockdown or overexpression (Fig. 5b and Supplementary Fig. 5a). Considering that recent studies have reported that PTMs regulated by SIRT5 affect enzyme activity strongly[25,27,32], we then measured the activities of GLS, GOT2, and GLUD1, which have many acylation sites across different species (Supplementary Table 3)[21–25,33,34]. Interestingly, suppression of SIRT5 in HCT116 and LoVo cells caused significant inhibition of GLUD1 activity, by 40% and 30%, respectively (Fig. 5c), which might result in the accumulation of the upstream substrate (glutamate), and deficiency of the downstream product (α-KG), supporting

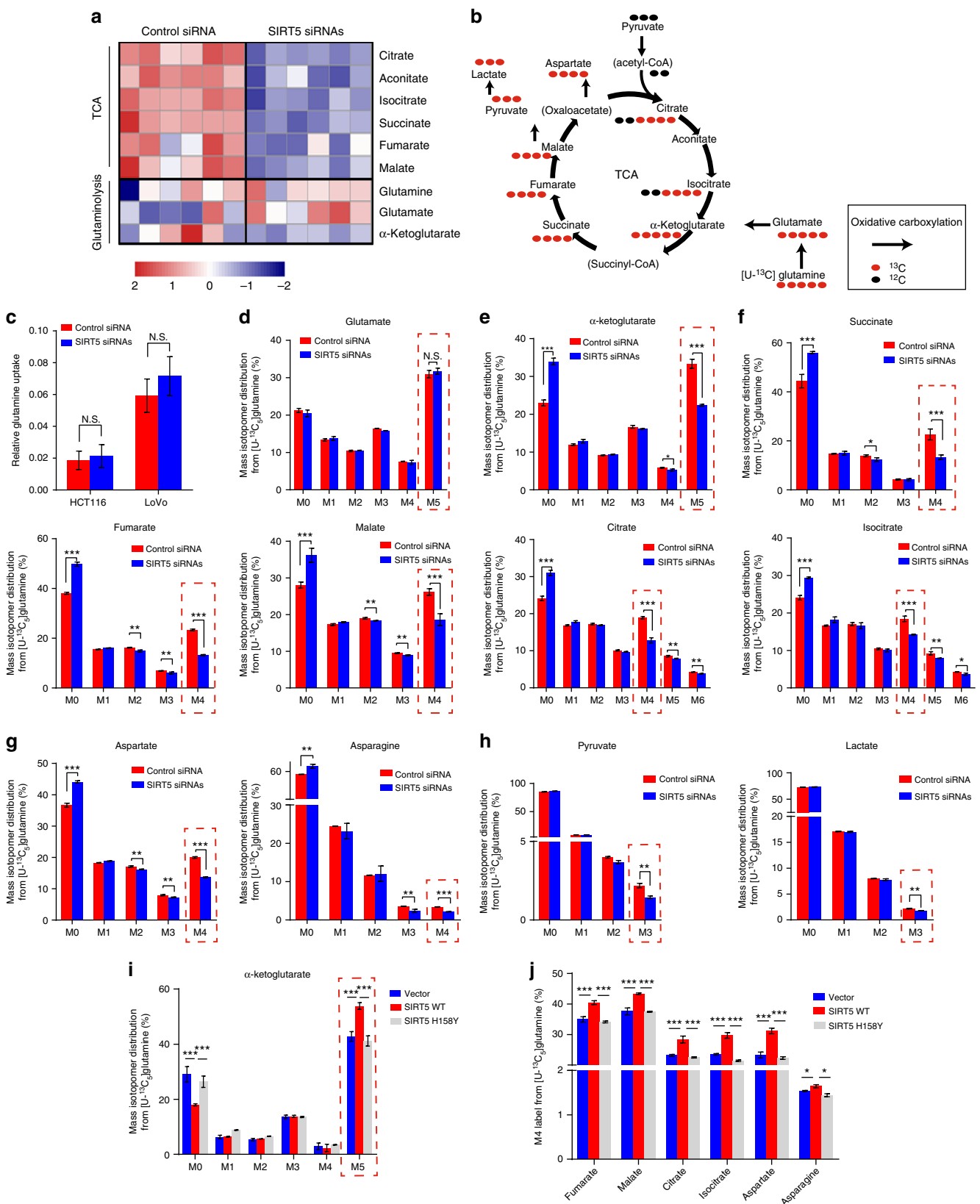

the GC-MS results in Fig. 4a. Moreover, the activity of GLUD1 (Fig. 5d), but not GLS (Fig. 5e) or GOT2 (Fig. 5f and Supplementary Fig. 5b), was significantly increased in SIRT5 WT-overexpressing cell lines. Conversely, the mutant showed completely abolished SIRT5-mediated GLUD1 activation. These data supported the view that the major step of glutamine metabolism regulated by SIRT5 is the glutamate-to-α-KG conversion conducted by GLUD1.

To determine whether the conversion from glutamate to α-KG by GLUD1 is required for the pro-survival effect of SIRT5, we tested the functional relevance of downstream metabolites of glutaminolysis in SIRT5-induced CRC proliferation. First, we analyzed the proliferation rate of cells in glutamine-free medium. Compared with complete media, glutamine deprivation reduced growth dramatically in both HCT116 and LoVo cells (Supplementary Fig. 5c). Next, the addition of the outputs of glutaminolysis, including dimethyl-glutamate (DM glutamate; a cell permeable form of glutamate), dimethyl-α-KG (DM α-KG, a cell permeable form of α-KG), or NEAA mixture (aspartate and asparagine, which were upregulated robustly upon SIRT5 overexpressed) increased the survival of CRC cells, confirming that glutaminolysis plays a critical role in CRC cells' proliferation (Fig. 5g). Notably, we demonstrated that only glutamate, but not other metabolites, restored the increased growth specifically induced by SIRT5 (Fig. 5g). These results supported the view that the glutamate-to-α-KG conversion mediates the observed effects on SIRT5-driven CRC proliferation.

As an important enzyme in glutamine metabolism, GLUD1 contributes to replenishment of the TCA cycle and cell viability. We then detected whether GLUD1 mediated the pro-proliferation effect of SIRT5. Vector control and SIRT5 WT-transfected cells were treated with a specific GLUD1 siRNA (Fig. 5h, i). Suppression of GLUD1 indeed blocked SIRT5-induced proliferation of CRC cells. In contrast, treatment with the pan-transaminases inhibitor AOA did not completely abolish SIRT5-mediated growth (Supplementary Fig. 5d). Considering that α-KG can be converted from glutamate by GLUD1 and that it connects the TCA cycle with glutamine metabolism, we speculated that decreased α-KG might be responsible for the inhibited proliferation upon SIRT5 knockdown. As expected, supplementation of cells with DM α-KG (1 mM) rescued the decreased proliferation in cancer cells after SIRT5 silencing (Fig. 5j), and also reduced the level of the apoptosis marker, cleaved PARP, which was induced by SIRT5 knockdown (Fig. 5k).

Based on these findings, we proposed that SIRT5 promotes CRC growth through enforced glutamine metabolism by increasing GLUD1 activity rather than upregulating its protein level.

**SIRT5 activates GLUD1 by mediating its deglutarylation.** To identify the molecular mechanism by which SIRT5 regulates GLUD1 activity, we studied the localization of both proteins using immunofluorescence. As shown in Fig. 6a, b, the green fluorescence representing FLAG-SIRT5 was superimposed with the red fluorescence representing GLUD1. The extremely similar fluorescence intensity line profiles of FLAG and GLUD1 in Fig. 6a, b also suggested a strong co-localization between GLUD1 and SIRT5 (Fig. 6c and Supplementary Fig. 6a, respectively). This physical interaction between FLAG-SIRT5 and endogenous GLUD1 was further confirmed using a co-immunoprecipitation assay. Intriguingly, the SIRT5 mutant did not show impaired binding with GLUD1 (Fig. 6d, e), implying that the catalytic domain of SIRT5 is not required for the GLUD1-SIRT5 interaction. Additionally, the association of endogenous GLUD1 with endogenous SIRT5 was detected in both HCT116 and LoVo cells (Fig. 6f, g). Using a glutathione S-transferase (GST) pull-down assay with recombinant GST-GLUD1 and His-SIRT5, we observed a specific interaction between in vitro-translated GLUD1 and SIRT5. No precipitate was detected when His-SIRT5 was transfected alone (Fig. 6h). In agreement with the observed co-localization in CRC cell lines, we also confirmed the strong spatial overlap between GLUD1 and SIRT5 in CRC tissues (Supplementary Fig. 6b). The above data indicated that GLUD1 interacts directly with SIRT5 in CRC.

Given that SIRT5 activates GLUD1 in a deacylation-dependent manner, we next asked whether this interaction could affect the lysine acylation level of GLUD1. Interestingly, the lysine glutarylation level of GLUD1 was decreased by 80% when SIRT5 was overexpressed (Fig. 6i, j), while the succinylation or malonylation of GLUD1 remained largely unaffected (Supplementary Fig. 6c). Additionally, overexpression of the catalytically inactive mutant led to hyperglutarylation of GLUD1 when compared with the SIRT5 WT vector (Fig. 6i, j). By contrast, we detected an increased glutarylation level of GLUD1 in SIRT5-depleted HCT116 cells (Fig. 6k). Next, we examined whether lysine glutarylation or other acylation modifications, such as succinylation or malonylation, would alter the activity of GLUD1. Acyl-CoA could serve as the donor molecule for the lysine acylation modification[25]. The immunoprecipitated hemagglutinin (HA)-tagged GLUD1 was incubated with acyl-CoA in vitro followed by determination of its enzyme activity. As expected, glutaryl-CoA incubation decreased the activity of GLUD1 significantly (Fig. 6l). However, this was not observed using succinyl- or malonyl-CoA as substrates for the succinylation or malonylation reactions (Supplementary Fig. 6d). These findings supported the view that glutarylation of GLUD1 might negatively regulate its activation.

**Fig. 4** SIRT5 enhances glutamine-driven TCA cycle metabolite abundances in colorectal cancer cells. **a** Heat map representing significantly different metabolites after SIRT5 deletion in HCT116 cells. Blue (red) indicates the relative down (up) regulation levels of TCA cycle and glutaminolysis intermediates compared with cells treated with the control siRNA; $n = 6$. **b** Schematic model of glutamine metabolism in cancer cells. Red circles represent carbons derived from [U-$^{13}$C$_5$] glutamine, and black circles are unlabeled. The black arrows indicate oxidative carboxylation flux from glutamine. **c** Glutamine uptake was determined in HCT116 and LoVo cells. At 48 h post transfection with SIRT5 siRNAs, the cells were placed in fresh medium. Metabolite levels were measured after 9 h of culture and normalized to the cell number. The results were normalized to the control siRNA. Data are the mean ± SD of five independent samples. Student's t-test. N.S. = not significant for the indicated comparison. **d–h** Mass isotopologue distributions of glutamate (**d**); TCA cycle metabolites including α-KG (**e**), succinate, fumarate, malate, citrate, and isocitrate (**f**); glutamine-derived aspartate and asparagine (**g**); and pyruvate and lactate (**h**) in LoVo cells treated with the control siRNA or SIRT5 siRNAs. Cells were cultured in [U-$^{13}$C$_5$] glutamine for 24 h before metabolites extraction and GC-MS analysis; $n = 3$, data in **d–h** are shown as the mean ± SD. Student's t-test. *$P < 0.05$, **$P < 0.01$, ***$P < 0.001$. N.S. = not significant for the indicated comparison. **i** LoVo cells stably expressing control vector, SIRT5 WT, and SIRT5 H158Y were cultured with [U-$^{13}$C$_5$] glutamine for 24 h before metabolite extraction and GC-MS analysis. Mass isotopologues of α-KG were identified by GC-MS. **j** The fraction of m + 4 fumarate, malate, citrate, isocitrate, aspartate, and asparagine in LoVo cells stably expressing the control vector, SIRT5 WT, and SIRT5 H158Y were measured by GC-MS; $n = 3$. Data in **i** and **j** are shown as the mean ± SD. ANOVA with Tukey's test. *$P < 0.05$, ***$P < 0.001$

We further probed SIRT5-dependent deglutarylation sites in GLUD1. As shown in a previous proteomic study (Supplementary Table 3), there are nine different lysine residues in GLUD1 that could potentially be modified by glutarylation. Among these, lysine 399, lysine 503, and lysine 545 are conserved in GLUD1 orthologs from humans to *Drosophila melanogaster*, indicating that these residues may be critical to some evolutionarily conserved function of GLUD1 (Supplementary Fig. 6e). The lysine (K) to arginine (R) mutation retains a positive charge and is often utilized as a deacylated mimetic. Therefore, we generated three plasmids encoding mutant HA-tagged GLUD1, in which lysine 399, lysine 503, and lysine 545 residues were substituted by

R, respectively. Ectopically expressed wild-type GLUD1, and the K399R, K503R, and K545R mutants, were transfected into HCT116 cells, followed by *SIRT5* knockdown. The glutarylation of GLUD1 was analyzed by western blotting. We found that the K545R mutation resulted in a significant reduction in glutarylation (Fig. 6m). Notably, SIRT5 suppression increased the glutarylation levels of wild-type GLUD1, and the K399R and K503R mutants, but not the K545R mutant, suggesting that GLUD1 was glutarylated in a SIRT5-dependent manner on lysine 545 (Fig. 6m). Consistent with this, K545R mutant GLUD1 did not respond to SIRT5-mediated regulation of enzyme activity (Fig. 6n), indicating that K545 in GLUD1 is a major glutarylation

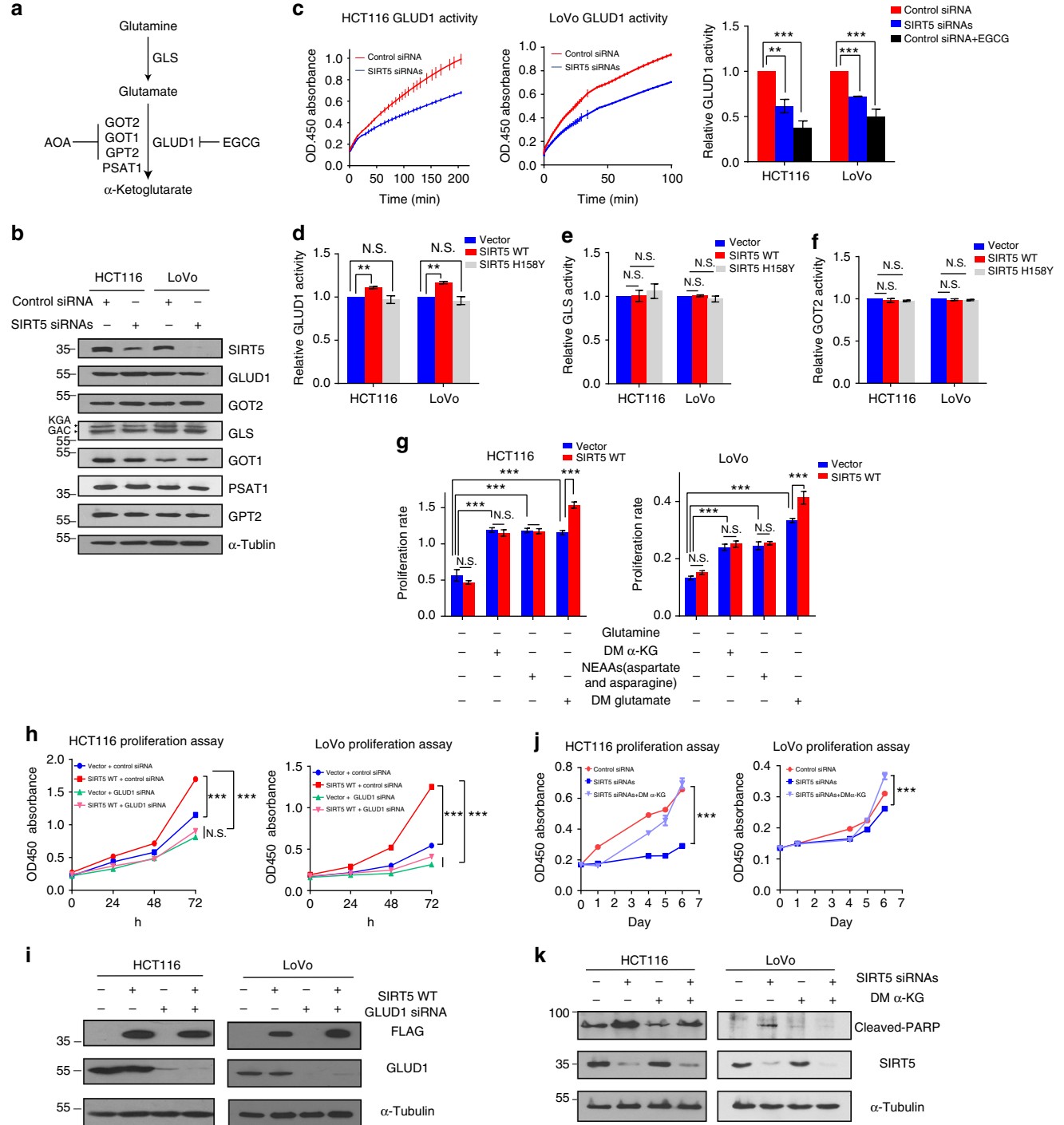

target of SIRT5. Taken together, these results suggested that ectopic expression of SIRT5 reduces glutarylation of GLUD1, which leads to its subsequent activation.

**Oncogenic role of SIRT5 in vivo**. To address the oncogenic role of SIRT5 in vivo, we established a xenograft tumor model in nude mice with HCT116 cells stably expressing control vector, SIRT5 WT, and SIRT5 H158Y. Tumor growth was evaluated and at the end of the experiment, tissue sections taken from the xenografts were subjected to GLUD1 enzyme activity analysis. Over-expression of SIRT5 WT accelerated CRC tumorigenesis significantly (Fig. 7a–c). On average, the SIRT5 WT-overexpressing tumors were approximately six times heavier than those from the control and mutant groups (Fig. 7d). Moreover, the GLUD1 activity in the SIRT5 WT tumors was increased by 38% compared with that in the control tumors, while it remained similar between SIRT5 H158Y- and control plasmid-treated tumors (Fig. 7e). Western blotting confirmed the overexpression of SIRT5 WT and SIRT5 H158Y in the xenograft tumor lysates, and we did not observe a significant change in GLUD1 protein level (Fig. 7f). Furthermore, we examined the function of GLUD1 in the tumorigenic properties of SIRT5 in vivo. Remarkably, the xenograft study showed delayed tumor growth, as well as decreased tumor size and mass, when GLUD1 was inhibited in SIRT5-overexpressing tumors (Fig. 7g–i), which suggested that GLUD1 knockdown abolished SIRT5-induced tumor growth significantly in vivo. Conversely, we found reduced CRC tumorigenesis, and decreased tumor volume and weight in the SIRT5 knockdown xenograft mice (Supplementary Fig. 7a–d). Consistent with our previous in vitro results, GC-MS analysis of tumor lysates also revealed that SIRT5 silencing resulted in a significant down-regulation of TCA cycle metabolites, including α-KG, succinate, fumarate, malate, citrate, and isocitrate (Supplementary Fig. 7e and 7f). Thus, these results recapitulated our cellular studies and supported strongly the model that SIRT5 contributes to the malignant phenotype of CRC by activating GLUD1 and subsequently enhancing TCA cycle flux.

## Discussion

A common characteristic of cancer metabolism is the ability to acquire and utilize nutrients to satisfy the demands of rapid proliferation[35]. In this study, we demonstrated a crucial role of SIRT5 in CRC metabolic reprogramming. We found that over-expression of SIRT5 promotes glutamine anabolic metabolism by activating GLUD1 in a deglutarylation-dependent manner, and is associated with CRC cell proliferation, survival, and xenograft tumor growth (Fig. 7j).

We observed that silencing of SIRT5 suppressed CRC cell proliferation dramatically by inducing apoptosis and cell cycle arrest. In xenograft mouse models, overexpression of SIRT5 promoted tumor growth significantly. Consistent with this in vitro and in vivo evidence, SIRT5 was upregulated extensively in CRC samples compared with their adjacent normal tissues, which was validated in the published GEO data sets. The protein level of SIRT5 is associated with tumor size, lymph node invasion, AJCC staging, and overall survival of patients with CRC. These findings support an oncogenic role of SIRT5, making it a promising biomarker of CRC.

Enhanced glutamine metabolism promotes macromolecular biosynthesis in malignant cells. In the present study, we demonstrated that glutamine metabolism is regulated by SIRT5 at the post-translational level in CRC. Our results showed that SIRT5 silencing blocked the formation of α-KG from glutamate, which in turn inhibited glutamine-derived metabolites from entering into the TCA cycle, consequently reducing precursors for anabolic biosynthesis. This process is essential to the colon cancer phenotype, but is dispensable in non-transformed cells[13]. Given that SIRT5 is found in relatively low levels in normal colonic mucosa and FHC cells, our findings highlight a tumor-specific metabolic vulnerability with a potential therapeutic application. It has been reported that the anticancer activity of glutaminolysis inhibition relies on the caspase-dependent apoptosis pathway[17], which is consistent with our results that blockage of SIRT5 induces cancer cell apoptosis triggered by caspase activation. Additionally, given the low abundance of aspartate in human plasma, and the lack of aspartate transport across the plasma membrane for aspartate uptake, cancer cells are generally dependent on de novo aspartate synthesis[36,37]. Interestingly, we revealed that SIRT5 directs glutamine-derived carbons into aspartate and asparagine synthesis through glutaminolysis, which is indispensable for de novo purine nucleotide and pyrimidine nucleotide synthesis in proliferating cells[38,39]. Therefore, our results suggested that SIRT5 represents a potential therapeutic value for the selective targeting of CRC, especially in cases with increased glutamine metabolism.

It is reported that GLUD1 catalyzes the reversible deamination of glutamate to produce α-KG in the liver[40]. However, in cancer, it operates mainly in the direction of α-KG formation[8], highlighting a crucial role of GLUD1 in α-KG synthesis in cancer. We identified GLUD1 as a major target of SIRT5 in promoting CRC glutaminolysis. We also showed that GLUD1 is critical and sufficient for SIRT5-mediated cancer progression, both in vivo and in vitro. Moreover, a recent report revealed that GLUD1 is an important regulator of redox homeostasis in cancer cells by controlling the intracellular levels of α-KG and its subsequent

**Fig. 5** SIRT5 supports colorectal cancer growth by promoting glutamine metabolism via increased GLUD1 enzyme activity. **a** A diagram showing the enzymes involved in glutamine metabolism and the inhibitors used in this study. **b** Expression of GLUD1, GOT1/2, PSAT1, GPT2, and GLS (including kidney-type glutaminase (KGA isoform), and glutaminase C (GAC isoform)) upon SIRT5 knockdown. **c** GLUD1 enzyme activity was determined upon SIRT5 knockdown in HCT116 and LoVo cells. Left, representative images (n = 3). Right, quantification of GLUD1 activity. GLUD1 inhibitor epigallocatechin gallate (EGCG; 20 μM) was used as a positive control. **d–f** GLUD1 (**d**), GLS (**e**), and GOT2 (**f**) enzyme activities were determined in HCT116 and LoVo cells stably expressing the control vector, SIRT5 WT, or SIRT5 H158Y, respectively; n = 3. **g** Proliferation rate of HCT116 and LoVo cells stably expressing the control vector or SIRT5 WT plasmid in different culture conditions. DM glutamate (10 mM), DM α-KG (1 mM), and non-essential amino acids (NEAAs; 0.1 mM aspartate and asparagine) were added to glutamine-free medium, respectively; n = 4. **h** CCK-8 assays of HCT116 and LoVo cells stably expressing the control vector or SIRT5 WT treated with/without the specific GLUD1 siRNA; n = 5. **i** GLUD1 was knocked down in HCT116 and LoVo cells stably expressing the control vector or SIRT5 WT vector. Protein levels were assessed by western blotting. **j** Growth curves of HCT116 and LoVo cells after transfection of control siRNA (red line) and SIRT5 siRNAs (blue line) under the indicated conditions. Cells were cultured in standard media, and DM α-KG (1 mM) was added to media as indicated. The OD 450 was measured for 6 consecutive days using the CCK-8 assay; n = 5. **k** HCT116 and LoVo cells were transfected with control siRNA or SIRT5 siRNAs, and then cultured in standard media with DM α-KG (1 mM). The levels of cleaved PARP were detected by western blotting. Data in **c–j** are presented as the mean ± SD. P values were calculated by ANOVA. **P < 0.01, ***P < 0.001, N.S. = not significant for the indicated comparison

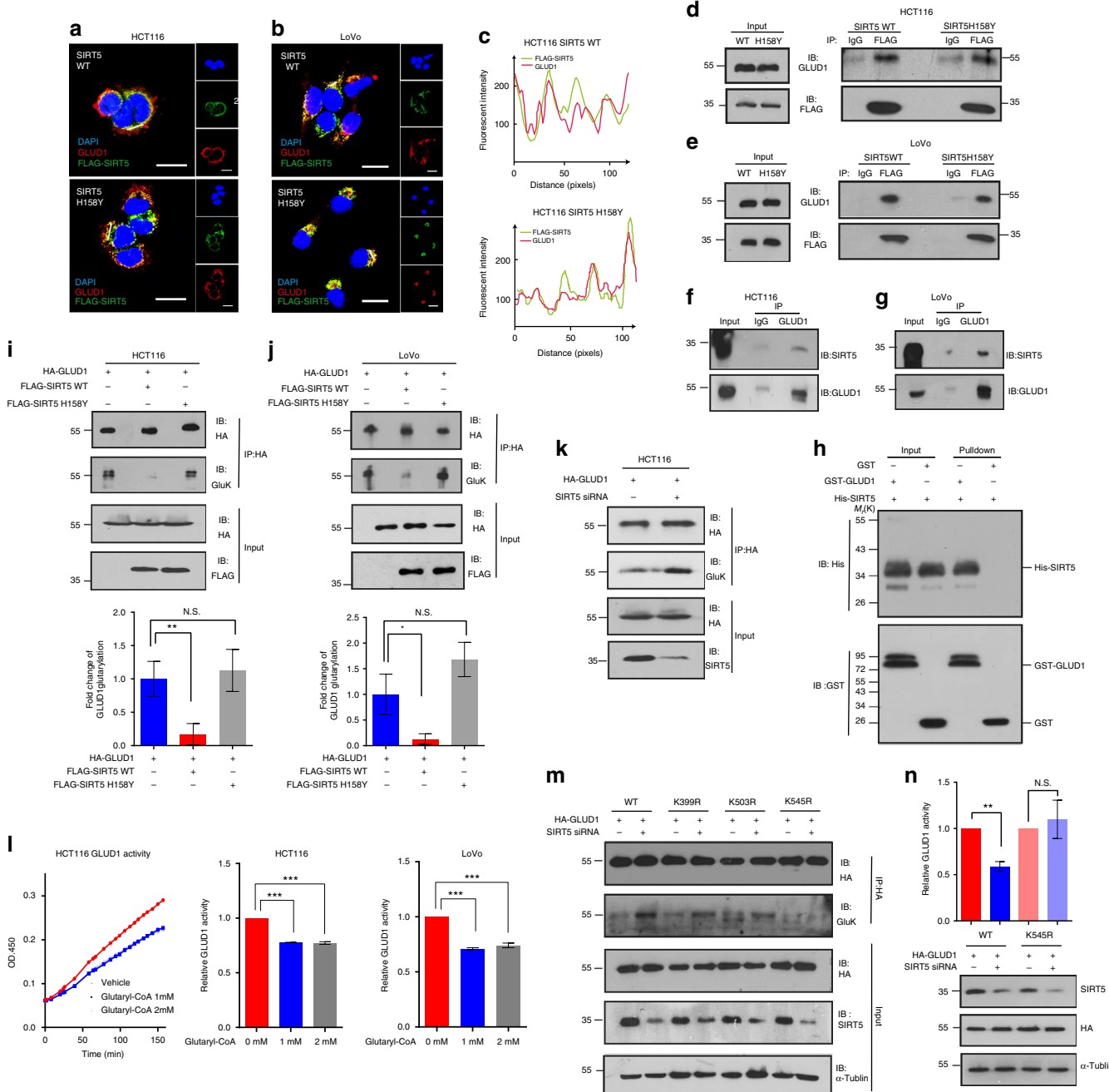

**Fig. 6** The direct interaction between GLUD1 and SIRT5 causes hypoglutarylation of GLUD1. **a**, **b** HCT116 (**a**) and LoVo (**b**) cells were immunostained for FLAG-SIRT5 WT/H158Y (in green) and GLUD1 (in red); yellow in the merged magnified images (left) indicates the co-localization. Scale bars indicate 20 μm. **c** Fluorescence intensity of FLAG-SIRT5 WT/H158Y (green line) and GLUD1 (red line) traced along the white line in HCT116 using the line profiling function of the ImageJ software. **d**, **e** FLAG-SIRT5 WT/H158Y were immunoprecipitated with anti-FLAG antibody and followed by western blotting with an anti-GLUD1 antibody in HCT116 (**d**) and LoVo (**e**) cells. **f**, **g** The interaction between endogenous GLUD1 and SIRT5 in HCT116 (**f**) and LoVo (**g**) cells. **h** GST pull-down assays revealed the interaction between SIRT5 and GLUD1. In vitro-translated SIRT5 was incubated GST fusion of GLUD1 or GST, and analyzed by western blotting. The bound SIRT5 is indicated by an arrow on the right. **i**, **j** Exogenous GLUD1 proteins were purified in HCT116 (**i**) and LoVo (**j**) cells expressing the control vector, SIRT5 WT, and SIRT5 H158Y; and the glutarylation (GluK) levels of GLUD1 were determined by western blotting. Integrated density values were calculated using the ImageJ software. **k** Exogenous GLUD1 was purified upon *SIRT5* knockdown, and the GluK level of GLUD1 was determined. **l** HA-tagged GLUD1 proteins were purified and incubated with different concentrations of glutaryl-CoA (0, 1, and 2 mM) at 37 °C for 60 min. The GLUD1 activity was determined. Left, representative images (*n* = 6). Right, quantification of GLUD1 activity. **m** Wild-type GLUD1, K399R, K503R, and K545R mutants were transfected into HCT116 cells followed by *SIRT5* knockdown. The GLUD1 was immunoprecipitated and the level of GluK was determined. **n** HCT116 cells expressing HA-tagged GLUD1 WT/K545R mutant were treated with or without *SIRT5* siRNAs. The GLUD1 activity was measured and normalized against the protein levels. The results in **i–n** are the mean ± SD of three independent experiments. *P* values were calculated by ANOVA with Tukey's test. \**P* < 0.05, \*\**P* < 0.01, \*\*\**P* < 0.001, N.S. = not significant for the indicated comparison

metabolite fumarate[41]. Additionally, the conversion of glutamate to α-KG catalyzed by GLUD1 is also coupled with NADPH production, which is required for redox control and cancer cell proliferation[42]. These results are consistent with the protective effect of SIRT5 against cellular oxidative stress[27,43].

Lysine glutarylation is an evolutionarily conserved PTM that is enriched on diverse metabolic enzymes[25]. However, how lysine deglutarylation affects cancer metabolism remains poorly understood. In this study, we detected an interaction between SIRT5 and GLUD1, leading to deglutarylation of K545, which activated GLUD1 in CRC. In fact, other PTMs, including acetylation[44] and ADP-ribosylation[45], have also been found to regulate GLUD1 function in cells. However, based on the negatively charged nature and the large size of the modification, it is very

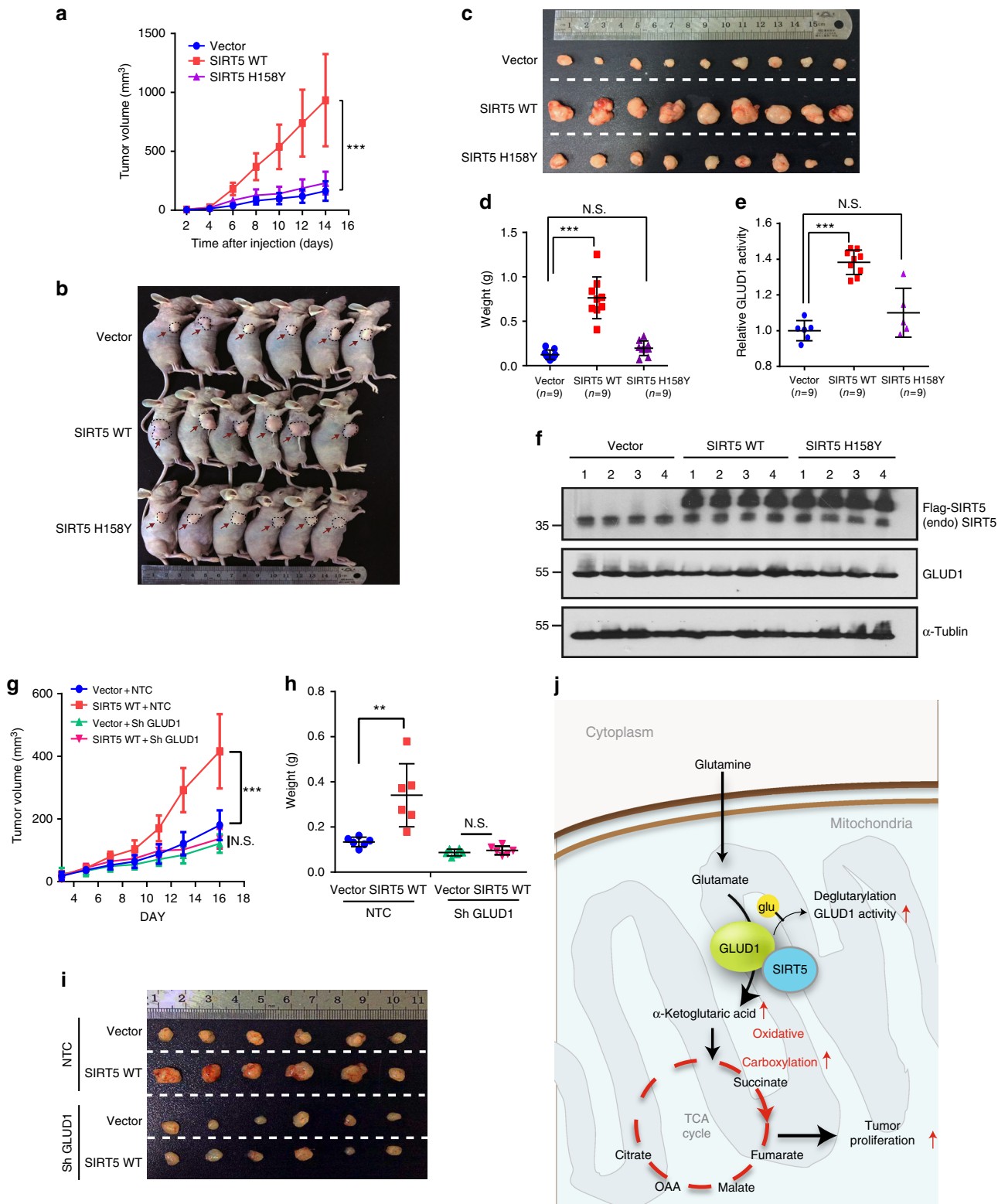

likely that glutarylation would have a more profound impact on protein structure and function compared with the other modifications[25,46]. Our results revealed that K545 in GLUD1 is a major glutarylation target of SIRT5. Based on the crystal structure (PDB ID: 1L1F), K545 is adjacent to the regulatory binding domain in GLUD1 (Supplementary Fig. 6f). The regulatory domain is situated near the pivot helix between adjacent protomers. An activator could bind to the regulatory domain and hasten the opening of the catalytic cleft that leads to activation of GLUD1[47,48]. Although further research is required to clarify the detailed mechanism, these findings extend our understanding of the biological function of PTMs in the regulation of tumor glutaminolysis.

In summary, our findings revealed that SIRT5 is a regulator of cancer glutaminolysis via its activation of GLUD1 in a deglutarylation-dependent manner, suggesting SIRT5 as a promising anti-CRC target for the selective killing of cancer cells.

## Methods

**Patient specimens**. The collection of histologically confirmed CRC tissues and adjacent non-tumor tissues was approved by the ethics committee of Shanghai Jiao Tong University School of Medicine (Shanghai, China), and written informed consent was obtained from the enrolled patients. The relevant clinical and histopathological data provided to the researchers were anonymized. All the research was carried out in accordance with the provisions of the Helsinki Declaration of 1975.

**Cell culture**. CRC cells HCT116 and LoVo were obtained from American Type Culture Collection and grown in McCoy's 5A and Roswell Park Memorial Institute (RPMI) 1640 (Gibco BRL, Grand Island, NY, USA) supplemented with 10% fetal bovine serum (FBS; Gibco BRL), respectively. The cell lines were tested for mycoplasma contamination before used to ensure that they were mycoplasma-free. For glutamine deprivation, cells were seeded overnight, briefly washed with phosphate-buffered saline (PBS; Hyclone, South Logan, UT, USA) and then transferred into glutamine-free RPMI 1640 (Gibco BRL, Gaithersburg, MD, USA) supplemented with 10% of dialyzed FBS (Biological Industries, Kibbutz Beit Haemek, Israel). The following chemicals were added into the culture media in this study: DM-α-KG (catalog #349631); aspartate (catalog #A8949); asparagine (catalog #4159); pan-transaminases inhibitor AOA (catalog #C13408); and GLUD1 inhibitor ECGC (catalog #4143) were purchased from Sigma-Aldrich (St. Louis, MO, USA). DM glutamate (catalog #D3305) was obtained from Tokyo Chemical Industry (Tokyo, Japan). Cell lines used in the study were grown in a humidified 5% $CO_2$ atmosphere at 37 °C.

**RNA interference**. SiRNAs specifically targeting SIRT5 and GLUD1 were purchased from GenePharma (Shanghai, China) using the following sequences: SIRT5: siRNA-1, 5′-GCUGGAGGUUAUUGGGAGAATT-3′; siRNA-2, 5′-GUGGCUGA-GAAUUACAAGATT-3′. These siRNAs were used as a pool for siRNA transfection. GLUD1: siRNA, 5′-GCGUUCUGCCAGGCAAAUUTT-3′; HCT116 and LoVo cells were seeded at 30% confluence in six-well plates overnight before transfection, and then transfected with 50 nM siRNA using Dharma FECT 1 transfection reagent (Dharmacon, Lafayette, CO, USA), according to the manufacturer's instructions. A nonspecific siRNA (GenePharma) was used as a negative control.

**Cell proliferation and clonogenic assay**. Cell numbers were measured spectrophotometrically using a CCK-8 (Dojindo, Kumamoto, Japan) according to the

manufacturer's instructions. In Fig. 5g, cells were seeded onto 96-well culture plates overnight. The next day, CCK-8 solution was added and the absorbance at 450 nm was determined ($OD_{450}$ absorbance DAY0). Media was changed according to different culture conditions following 4 days of incubation. At the end of the study, the absorbance at 450 nm was measured ($OD_{450}$ absorbance DAY4). The proliferation rate was calculated as follows: $OD_{450}$ absorbance DAY4-blank/$OD_{450}$ absorbance DAY0-blank. For the clonogenic assay, 500 cells were seeded in 6-well plates, and cultured for 10 days. Colonies were fixed with 4% paraformaldehyde, and then stained with 0.5% crystal violet. Megascopic colonies were counted. Experiments were repeated at least three times.

**Flow cytometry assays**. The effect of SIRT5 on apoptosis and cell cycle progression was analyzed by flow cytometry. HCT116 and LoVo cells were collected after *SIRT5* siRNA transfection and analyzed using an FITC Annexin V Apoptosis Detection Kit I (BD Pharmingen, San Diego, CA, USA) according to the manufacturer's instructions. For the cell cycle assay, cells were harvested and fixed in ice cold 70% ethanol, followed by staining with 50 μg/mL propidium iodide containing 20 μg/mL RNase (DNase-free). Stained cells were analyzed by flow cytometry. The percentage of cells in distinct phases was recognized according to the content of DNA: G1, S (DNA synthesis phase), G2, and M (mitosis) phases. All experiments were performed in triplicate and repeated at least three times.

**Stable cell line generation**. To establish stable SIRT5-expressing cells, control vector, and FLAG-SIRT5, H158Y plasmids were transfected into HCT116 and LoVo using the FuGENE HD Transfection Reagent (Promega, Madison, WI, USA). Non-target control (NTC) short hairpin RNA (shRNA) or SIRT5 shRNA were transfected into HCT116 to generate cells with stable knockdown of SIRT5. After 48 h of transfection, cells were selected in medium containing G418 (1.5 mg/mL, MP Biomedicals, Irvine, CA, USA) for 30 days. The stable colonies resistant to G418 were selected and confirmed by western blotting, and then further cultured in medium supplemented with adequate amounts of antibiotics.

**Western blotting**. Cells or tissues were lysed with lysis buffer (50 mM Tris 7.4, 150 mM NaCl, 1% NP-40, 0.5% sodium deoxycholate, and 0.1% SDS) supplemented with a protease inhibitor cocktail (Kangcheng, Shanghai, China). Lysates were separated by SDS-polyacrylamide gel electrophoresis (SDS-PAGE) and immunoblotted. The antibodies used were as follows: anti-SIRT5 (catalog #HPA022002, 1:2000, Sigma-Aldrich), anti-cleaved caspase 3 (catalog #9664, 1:1000), anti-cleaved caspase 8 (catalog #9496, 1:1000), anti-cleaved caspase 6 (catalog #9761, 1:1000), anti-cleaved PARP (catalog #5625, 1:1000), anti-γ-H2A.X (catalog #9718, 1:1000), anti-CDK4 (catalog #12790, 1:1000), anti-cyclin D1 (catalog #2978, 1:1000), anti-cyclin D3 (catalog #2936, 1:1000), anti-cyclin A2 (catalog #4656, 1:1000), anti-cyclinB1 (catalog #12231, 1:1000), anti-cyclin E1 (catalog #20808, 1:1000), anti-p21 (catalog #2947, 1:1000), and anti-α-tublin (catalog #2148, 1:2000) were purchased from Cell Signaling Technology (Danvers, MA, USA); anti-FLAG (catalog #F1804, 1:1000, Sigma-Aldrich); anti-HA (catalog #MMS-101P, 1:1000, Convance, Princeton, NJ, USA); anti-GLUD1 (catalog #14299-1-AP, 1:1000), anti-GOT2 (catalog #14800-1-AP, 1:1000), anti-GPT2 (catalog #16757-1-AP, 1:600), anti-GLS (catalog #12855-1-AP, 1:1000), anti-PSAT1 (catalog #10501-1-AP, 1:1000), and anti-GOT1 (catalog #14886-1-AP, 1:1000) were purchased from Proteintech (Chicago, IL, USA); anti-pan succinylation (catalog #PTM-401, 1:1000), anti-pan glutarylation (catalog #PTM-1151, 1:2000), and anti-pan malonylation (catalog #PTM-901, 1:1000) were obtained from PTM Biolabs (Hang-Zhou, China). Peroxidase-conjugated anti-rabbit and anti-mouse secondary antibodies (1:5000, Kangcheng) were used as secondary antibodies. Signals were detected using Western ECL Substrate (Thermo Scientific, Waltham, MA, USA). Three independent experiments were performed for each analysis. Original images of western blots are shown in Supplementary Fig. 8-9.

**Enzyme activity measurements**. The enzyme activity of GLUD1 and GOT were determined using a Glutamate Dehydrogenase Activity Colorimetric Assay Kit

**Fig. 7** Oncogenic role of SIRT5 and GLUD1 is vital for the tumorigenesis capacity of SIRT5 in vivo. **a** HCT116 cells stably expressing the control vector, SIRT5 WT, or SIRT5 H158Y were injected subcutaneously into nude mice (*n* = 9 for each group). Tumor volumes were measured at the indicated time points and the mean tumor volumes were calculated. Data are presented as the mean ± SD. **b–d** At the end of experiment, tumors from three groups were dissected, photographed (**b**, **c**), and weighed (**d**). Data are presented as the mean ± SD. **e**, **f** GLUD1 enzyme activities in tumor lysates derived from xenografts were measured (**e**). Data are presented as the mean ± SD. GLUD1 protein level and the overexpression of SIRT5 in the xenografts were confirmed by immunoblotting (**f**). **g–i** SIRT5-overexpressing LoVo cells infected with viruses expressing non-target control (NTC) short hairpin RNA (shRNA) or GLUD1 shRNA were injected subcutaneously into nude mice (*n* = 6 for each group). Tumor growth curves were constructed (**g**). Statistical analysis of tumor weight. Each dot represents the tumor mass from one mouse (**h**). Digital photograph of the dissected tumors (**i**). Data are presented as the mean ± SD. All *P* values were calculated by ANOVA with Tukey's test. **\*\****P* < 0.01, **\*\*\****P* < 0.001, N.S. = not significant for the indicated comparison. **j** Schematic model showing the suggested role of SIRT5 in the regulation of glutamine metabolism in CRC. SIRT5 directly interacts with GLUD1, which functions as a critical enzyme responsible for the formation of α-KG from glutamate. This interaction results in the deglutarylation and activation of GLUD1, leading to direct glutamine flux into the mitochondrial TCA cycle, thus promoting oxidative carboxylation of glutamine. The enhanced glutaminolysis supports CRC cell proliferation

(catalog #K729-100, BIoVision, Milpitas, CA, USA) and a Glutamate Oxaloacetate Transaminase Activity Assay Kit (catalog #K753-100, BIoVision), respectively. Briefly, $2 \times 10^6$ cells or 50 mg of tissues were homogenized, mixed with the assay buffer, and incubated for 2 min at 37 °C in the dark. The change in absorbance at 450 nm was measured every 5 s for 10 min at 37 °C with a microplate spectro-photometer (Thermo Fisher Scientific, Waltham, MA, USA) according to the manufacturer's instructions. GLS enzyme activity was measured by a Glutaminase Assay Kit (catalog #E-133, Biomedical Research Service Center, Buffalo, NY, USA) according to the manufacturer's instructions. Cells ($2 \times 10^6$) were homogenized, mixed with the glutamine solution, and incubated for 2 h at 37 °C. The assay buffer was then added to the samples, followed by incubating for another 1 h at 37 °C. The optical density (OD) at 492 nm was then measured using a microplate spectro-photometer. Each sample was analyzed in triplicate, and the enzyme activity was calculated following the manufacturer's instruction. Experiments were repeated at least three times.

**Metabolite-level measurements**. To analyze intracellular metabolites by GC-MS, HCT116 cells ($1 \times 10^7$/sample) or tumor tissues (50 mg) were quenched and metabolites were derivatized with methoxyamine (15 mg/mL in pyridine) for 90 min at 37 °C, and subsequently with Bis (trimethylsilyl) trifluoroacetamide (1% chlorotrimethylsilane) and 20 μl of n-hexane for 60 min at 70 °C. For stable isotope-tracing analysis, LoVo cells ($2 \times 10^6$/sample) were grown to 80% confluence in complete media. The media was replaced with glutamine-free RPMI 1640 (Gibco BRL) supplemented with 2 mM [U-$^{13}C_5$] glutamine or [1-$^{13}$C] gluta-mine(Cambridge, Isotope Laboratories, Andover, MA, USA) and 10% dialyzed FBS (Biological Industries) for 24 h. Cells were quenched and metabolites were derivatized with methoxyamine (15 mg/mL in pyridine) for 90 min at 37 °C, sub-sequently with N-(ter-Butyldimethylsilyl)-N-methyltrifluoroacetamide at 55 °C for 60 min.

Metabolomics instrumental analysis was performed on an Agilent 7890A gas chromatography system coupled to an Agilent 5975 C inert MSD system (Agilent Technologies Inc., CA, USA). A HP-5ms fused-silica capillary column (30 m × 0.25 mm × 0.25 μm; Agilent J&W Scientific, Folsom, CA, USA) was utilized to separate the derivatives. Mass spectra were collected from m/z 50–600 under the selected reaction monitoring mode. For the analysis of intracellular metabolites, PLS-DA was carried out to visualize the metabolic alterations among the experimental groups. Variable importance in the projection (VIP) ranks the overall contribution of each variable to the PLS-DA model. Significantly different metabolites were identified by a combination of two methods: $P < 0.05$ by t-tests, and VIP > 1 by PLS-DA. For stable isotope-tracing analysis, the measured distribution of mass isotopomers was corrected for the natural abundance of isotopes using the software IsoCor. Labeled metabolite data were expressed using the percentage of the total pool or relative ion abundances. Metabolite levels were yielded by normalizing them to the internal standard and cell number from parallel plates.

The amount of glutamine uptake by cells was determined using a Glutamine Assay Kit(catalog #EGLN-100, Bioassay Systems, Hayward, CA, USA). The culture media were incubated with the respective enzyme mix at room temperature for 40 min in the dark following the manufacturer's protocol. The absorbance was read at 565 nm with a microplate spectrophotometer (Thermo Fisher Scientific).

**Immunofluorescence**. For immunofluorescence of cultured cells, HCT116 and LoVo cells were stained using mouse monoclonal anti-FLAG (1:1200) and rabbit polyclonal anti-GLUD1 (1:200) antibodies, followed by the donkey anti-mouse Dylight 488 (1:400) and donkey anti-rabbit Dylight 594 (1:400) secondary anti-bodies (Thermo Fisher Scientific). For microarray immunofluorescence, the tissue sections were deparaffinized in xylene and rehydrated using a graded series of ethanol. All slides were treated with NaBH$_4$ to suppress tissue autofluorescence. The expression levels of SIRT5 were probed using primary antibodies (SIRT5, 1:100) according to the manufacturer's instructions. The mitochondria were pro-bed using an anti-mitochondria antibody (catalog #ab3298, 1:200). Secondary antibodies (Alexa 488-anti-rabbit, dilution 1:400, donkey anti-mouse Dylight 594, 1:400) were used. Nuclei were counterstained with 2-(4-amidinophenyl)-1H-indole-6-carboxamidine. Fluorescence was analyzed using a confocal laser-scanning microscope (Carl Zeiss, AG, Germany). The average fluorescence intensity was quantified using Zeiss digital image processing software, ZEN® (blue edition). Analysis of the clinicopathological features excluded a few patients because of missing data (in Supplementary Table 1).

**Quantitative real-time reverse transcription PCR**. Total RNA was extracted using the TRIzol reagent (Invitrogen, Carlsbad, CA, USA). A First Strand cDNA Synthesis Kit (Takara, Tokyo, Japan) was used for reverse transcription according to the manufacturer's protocol. Quantitative real-time PCR was performed using an ABI Prism 7900HT Sequence Detection System (Applied Biosystems, USA) with SYBR Premix Ex Taq II (Takara). Quantification was calculated using the $^{2-\Delta\Delta}$CT method and is presented as fold change normalized to that of 18S RNA. The sequences of primers used in this study (forward and reverse sequences for each gene): 18S: 5′-CGGACAGGATTGACAGATTGATAGC-3′, 5′-TGCCA-GAGTCTCGTTCGTTATCG-3′; SIRT3: 5′-CCCCAAGCCCTTTTTCACTTT-3′, 5′-CGACACTCTCTCAAGCCCA-3′; SIRT4: 5′-

ACCCTGAGAAGGTCAAAGAGTTAC-3′, 5′-TTCCCCACAATCCAAGCAC-3; and SIRT5: 5′-TATTAGAAAGCAGCCGTGGAGA-3′, 5′-CGCAT-CAGGGTTTGTCTGTAG-3′.

**Immunoprecipitation and GST pull-down experiments**. In HCT116 and LoVo cells stably expressing the control vector, SIRT5 WT vector, or SIRT5 H158Y, CO-IP and IP were performed using the Pierce Co-IP Kit (catalog #26149, Thermo Fisher Scientific). Eight micrograms of anti-FLAG antibody, anti-GLUD1 antibody, or normal immunoglobulin G were coupled to Amino Link Plus Coupling Resin according to the manufacturer's protocol. For cells expressing wild-type GLUD1, K399R, K503R, and K545R mutant, 10 μg of HA antibody was coupled to Amino Link Plus Coupling Resin. Cell lysates were pre-cleared using control agarose resin, and incubated with antibody-coupled resin with gentle mixing overnight at 4 °C. Immune complexes were eluted from the resin and analyzed by SDS-PAGE, fol-lowed by immunoblotting analysis. For the GST pull-down experiment, GST and GST-GLUD1 fusion proteins were expressed and purified according to the man-ufacturer's instructions (GE Healthcare, Little Chalfont, Buckinghamshire, UK). Equal amounts of GST or GST fusion proteins were mixed with glutathione-Sepharose 4B beads (GE Healthcare) in binding Buffer (150 mM NaCl, 25 mM HEPES, pH 7.6, and 1 mM dithiothreitol) for 2 h at 4 °C, and then the purified His-SIRT5 protein was added, followed by incubation for another 4–8 h at 4 °C. The pellets were washed by PBS three times, and identified by western blotting using anti-His and anti-GST antibodies (1:2000, CMCTAG, San Diego, CA, USA).

**In vivo models**. Nude mice (nu/nu, male, 5 weeks old) were weighed, sorted according to the weight, and allocated to experimental groups using the random number table. Mice were injected subcutaneously with CRC cells ($5 \times 10^6$ cells) stably expressing the control vector, SIRT5 WT, SIRT5 H158Y, or SIRT5 WT, with or without GLUD1 knockdown. For the loss-of-function experiments, HCT116 cells stably expressing the NTC shRNA or SIRT5 shRNA ($5 \times 10^6$ cells) were injected subcutaneously into 5-week-old nude mice. The diameters of tumors were measured using calipers every 2 or 3 days. The tumor volumes were calculated using the formula: (shortest diameter)$^2$ × (longest diameter) × 0.5. At the end point, the tumors were dissected and analyzed. All animal studies were conducted in accordance with the guidelines published in the Animal Ethics Committee of Renji Hospital Shanghai Jiao Tong University School of Medicine.

**Statistical analysis**. Data are presented as the mean ± SD. Comparisons of only two conditions were performed using Student's t-test (two-tailed). For multiple comparisons, analysis of variance was used for the statistical analysis. Immuno-histochemistry results were analyzed using the $\chi^2$-test in the Statistical Package for Social Sciences software. $P$ values < 0.05 were accepted as statistically significant.

**Data availability**. The metabolites data are available in figshare with the identifier (data DOI: 10.6084/m9.figshare.5731485). The authors declare that all the other data supporting the findings of this study are available within the article and its Supplementary Information files and from the corresponding author on reasonable request.

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

## Acknowledgements

We would like to thank Professor Chen Yang for her valuable technical assistance with metabolic assay. We thank Dr Yun Cui and Dr Qi Miao for their help with the IHC staining. We also thank Dr Ying-Chao Wang for his help with flow cytometry analysis. This work was supported by the National Natural Science Foundation of China (grant numbers 81572303 to Y.-X.C.; 81530072 and 81421001 to J.-Y.F.; and 81001070 to D.-F.S.), the National Key Technology R&D Program (grant number 2014BAI09B05), the Shanghai Municipal Education Commission-Gaofeng Clinical Medicine Grant (grant number 20152210), and the Program of Shanghai Academic/Technology Research Leader (grant number 15XD1502600).

## Author contributions

Y.-Q.W., J.X. and Y.-X.C. designed the experiments, analyzed the data, and wrote the manuscript. Y.-Q.W., H.-L.W., J.T., L.-N.F. and T.-H.Z. performed the experiments. D.-F.S., Q.-Y.G. and J.-L.W. supplied critical materials to the study. J.-Y.F. reviewed and revised the manuscript. Y.-X.C. supervised the project. All the authors have read and approved the final version of manuscript.

## Additional information

**Competing interests:** The authors declare no competing financial interests.

