## [Peer Review File · Nature Communications]

Reviewers' comments:

Reviewer #1:

(Remarks to the Author):

This paper provide a novel mechanism by which SIRT5 can induce glutaminolysis and cell proliferation in colorectal cancer via the activation of GLUD1. The authors need to address the following issues to further improve the manuscript.

1. Figure 1. What are the potential mechanisms that lead to the up-regulation of SIRT5 in CRC?
2. Figure 2. It would be useful to validate the effect of SIRT5 on cell cycle progression by western blot analysis of cell cycle markers such as cyclin D1, p21, etc.
3. Figure 5. The authors should confirm the non-involvement of transaminases using inhibitors such as Aminooxyacetic acid (AOA).
4. Figure 6. How might the hypoglutarylation of GLUD1 leads to its activation? Can the authors further elaborate on this?
5. Figure 7. The in vivo evidence would be more convincing if the authors determine the levels of TCA cycle metabolites in the xenograft models.

Reviewer #2:(Remarks to the Author):

In this study, Wang et al describe a role for SIRT5 in sustaining TCA pools via control of glutaminolysis via deglutarylation of GDH. Overall, the evidence that SIRT5 supports cell proliferation is strong and has been shown using multiple strategies. However, evidence supporting the underlying mechanism is less convincing. Additional studies directed at the direct effects of SIRT5 on metabolism and its influence over metabolic flux would significantly strengthen the work.

MAJOR:

-Cell proliferation. Given that cellular growth changes metabolic requirements, how much is growth rate playing a role in the isotope tracing effects observed? Faster growing cells (SIRT5 o/e) or slower growing cells (siSIRT5) would be expected to behave exactly as shown here (more/less glutamine metabolism), independent of SIRT5. This is a key point, and is not discussed. This is especially relevant given that the effects are subtle. Could differences in growth rate be driving the differences in metabolic labeling?

-Analyses of Isotope labeling. There are several problems with the way the isotope labeling data are presented and interpreted. First, it doesn't appear that the cells are in steady-state at the time points measured (Fig 4). The dramatic reduction in of isotope signal in the knock-down indicates this, even at the 3h-timepoint, which was subsequently used for over-expressor data. Further, the effect of SIRT5 on the dilution (wash-out) of isotope appears to be stronger than the labeling kinetics, which does not necessarily support their hypothesis. Ideally, the cells should be at a steady-state of growth to that fluxes can be interpreted. Additionally, changes in reductive carboxylation observed under SIRT5 overexpression are modest and do not seem to be enough to account for the significant changes in cell proliferation. Finally, all of the isotopologues should be shown (such as for the knock-down data), not just specific isotopes. At minimum, these should be included in the supplement to allow for interpretation of the data. Lastly, Fig. 4E doesn't appear to reflect 4D and 4F; where did this come

from?

-GLUD1 as a primary target. Chemical glutarylation of GLUD1 using glutaryl-CoA results in a slight reduction in its activity which cannot completely explain the significant lack of proliferation in absence of SIRT5. Blocking GDH with ECGC is sufficient to block all cell growth, independent of SIRT5, so while interesting, this reagent does not necessarily support a role for SIRT5 regulating GDH in this study. AIn the xenograft model increase in GLUD1 activity with SIRT5 overexpression is impressive but the authors do not show if there was an appreciable increase in GLUD1 expression in this model or is this increase in activity indeed due to SIRT5-dependent posttranslational modification. A detailed investigation of which sites on GLUD1 are important were not performed; identifying the sites of regulation would strengthen this argument.

MINOR:

-The paper is set-up with the concept and findings that SIRT5 is over-expressed and over-abundant in CRC, and correlates with poor prognosis. But then the authors move onto studies with SIRT5 silencing, and then make the conclusion that "inhibition of apoptosis and promotion of cell cycle progression could explain the observed SIRT5-induced proliferation of CRC cells". However, this is never directly tested. If possible, the authors should directly test their conclusions, rather than infer them from the opposing result (e.g. if knock-down leads to less growth and more apoptosis, then higher expression explains more growth and less apoptosis). The authors began to test some phenotypes directly in Fig. 3, but never tested cell cycle or apoptosis in their over-expressing lines. The inconsistencies in measurements from the over-expressers and knock-downs is puzzling.

-During the initial screen of the CRC tissue authors point out that the expression of SIRT5 is mainly cytoplasmic however later they attribute the mechanism of increased proliferation of CRC cell lines to changes in a post-translational modification, protein glutarylation, which is primarily mitochondrial on a protein GLUD1 which again is primarily mitochondrial. Although authors later show co-localization of GLUD1 and SIRT5 in the CRC cell lines how do the authors reconcile the difference in expression of SIRT5 in cells Vs CRC tissue?

Reviewer #3:

(Remarks to the Author):

The manuscript by Wang et al. describes the role of Sirtuin5 in malignant colorectal cancer through its activation of GLUD1, which converts glutamate to a-ketoglutarate. The authors begin by showing increased SIRT5 expression in human tissues through their own data and publicly available GEO datasets. To confirm biological relevance of SIRT5, the authors performed siRNA knock-down (KD) of SIRT5 in 2 cell lines and showed decreased proliferation, cell cycle arrest, and increased apoptosis in the KD cells. The converse experiment with overexpression of wild-type SIRT5 or catalytically inactive SIRT5 showed a growth improvement with wild-type SIRT5 overexpression. The authors also traced the metabolism of [U-13C5]glutamine in SIRT5 KD cells and showed a reduction in TCA cycle labeling from citrate independent of changes in glutamate labeling. The authors also worked to identify the mechanism of GLUD1 deglutarylation through direct interaction with SIRT5. Finally, the authors show robust tumor growth of SIRT5 wild-type overexpressing tumors compared to catalytically inactive SIRT5 tumors. Furthermore, this growth advantage of wild-type overexpressing SIRT5 is ablated with shRNA KD of GLUD1.

This manuscript is well organized and clearly explains the experimental designs and data interpretation. Some revisions are needed, however, before acceptance for publication.

Major points:

- To what extent does glutaminolysis extend to pyruvate and lactate labeling? Do the authors see pyruvate or lactate labeling from [U-13C5]glutamine?
- Following this point, M+5 citrate and m+3 malate, fumarate, and aspartate are used to indicate reductive glutamine metabolism. However, m+5 citrate can be formed through condensation of glutaminolysis-derived pyruvate AcCoA + m+3 oxaloacetate. Reductive glutamine metabolism to citrate, especially in the presence of high anaplerotic glutamine flux or high malic enzyme activity, is best assessed with [1-13C]glutamine.
- The authors see robust labeling of TCA cycle intermediates on these short time scales. It may present a more complete picture of total tracer incorporation to plot each intermediate's mole percent enrichment from [U-13C5]glutamine.
- Do the authors have a theory for why the conversion of glutamate to α -KG is required for the pro-survival effect of SIRT5 and not simply maintenance of the α -KG pool? Is glutamate limiting in SIRT5 overexpression cells (similar to how SIRT5 KD causes an increase in glutamate levels)?

Minor points:

- In plots of cell growth the y-axis is labeled as "proliferation rate." Is this raw cell number or an actual proliferation rate? If this is a calculated rate an equation should be included in the methods.
- Potential error in Supplementary Fig 2C: y-axis label of %m+5 does not seem possible given the time of cells in culture and compared to other Fig 4 graphs.
- Fig 5G should reflect that NEAAs added were only aspartate and asparagine.
- In figures with many t-tests, especially those where multiple bars are compared to a single control condition, the authors should use an ANOVA or at least correct for multiple comparisons within a single figure panel.

Responses to reviewers,

Dear reviewers: Thank you very much for your reviews of our manuscript. Based on your comments and requests, we have made extensive modifications to the original manuscript. Here, we have attached the revised manuscript in the formats of MS word, for your approval. Our point-by-point responses to your comments have also been summarized and enclosed as a separate document. A revised manuscript with the corrections marked in red is attached.

Reviewer #1: (Remarks to the Author):

This paper provide a novel mechanism by which SIRT5 can induce glutaminolysis and cell proliferation in colorectal cancer via the activation of GLUD1. The authors need to address the following issues to further improve the manuscript.

1. Figure 1. What are the potential mechanisms that lead to the up-regulation of SIRT5 in CRC?

Author response:

Thank you for your valuable suggestions.

[Editorial Note: Unpublished data redacted from Peer Review File as per Authorial request.]

2. Figure 2. It would be useful to validate the effect of SIRT5 on cell cycle progression by western blot analysis of cell cycle markers such as cyclin D1, p21, etc.

Author response:

Thank you for this excellent suggestion. Western blotting analysis was performed to confirm the effects of SIRT5 knockdown on both G2/M phase and S phase cell cycle arrest. Consistent with the flow cytometry, our results showed that depletion of SIRT5 resulted in a dramatic accumulation of cyclin E1 and cyclin A2, which was correlated with an arrest of the cell division cycle at the S phase checkpoint. In addition, we detected a reduction of the G1 phase regulators, including cyclin D1, cyclin D3, and CDK4, while the expression of cyclin B1 and p21 remained unchanged upon SIRT5 suppression (revised Figure 2G).

3. Figure 5. The authors should confirm the non-involvement of transaminases using inhibitors such as Aminooxyacetic acid (AOA).

Author response:

According to the reviewer's suggestion, the effect of aminooxyacetic acid (AOA) on SIRT5-induced proliferation was determined. HCT116 and LoVo cells stably expressing the control vector or SIRT5 WT plasmids were treated with/without the pan-transaminases inhibitor AOA (0.5 mM, Sigma #C13408). We found that ectopic expression of SIRT5 significantly promoted the growth of CRC cells ($P < 0.001$). Although AOA treatment suppressed the growth of SIRT5 WT transfected cells, the inhibition of transaminases could not completely abolish SIRT5-mediated cancer cell proliferation (revised Supplementary Figure 5D).

4. Figure 6. How might the hypoglutarylation of GLUD1 leads to its activation? Can the authors further elaborate on this?

Author response:

To probe the mechanism by which the hypoglutarylation of GLUD1 leads to its activation, we studied the SIRT5-dependent deglutarylation sites in GLUD1. There

are nine different lysine residues in GLUD1 that could potentially be modified by glutarylation, which were identified in a previous proteomic study (Supplementary Table 3). Among these, lysine 399, lysine 503, and lysine 545 are conserved in GLUD1 orthologs from humans to *Drosophila melanogaster*, indicating that these residues may be critical to some evolutionarily conserved function of GLUD1 (revised Supplementary Figure 6E). The lysine (K) to arginine (R) mutation retains a positive charge and often acts as a deacylated mimetic. Therefore, we generated three plasmids encoding mutant HA-tagged GLUD1, in which lysine 399, lysine 503, and lysine 545 residues were substituted by arginines (R), respectively. Ectopically expressed wild-type GLUD1, and the K399R, K503R, and K545R mutants were transfected into HCT116 cells, followed by SIRT5 knockdown. The glutarylation of GLUD1 was analyzed by western blotting. We found that the K545R mutation resulted in a significant reduction in glutarylation. Notably, SIRT5 suppression increased the glutarylation levels of wild-type GLUD1, and the K399R and K503R mutants, but not the K545R mutant, indicating that GLUD1 was glutarylated in a SIRT5-dependent manner on lysine 545 (revised Figure 6M). Consistent with this, K545R mutant GLUD1 displayed no response to SIRT5-mediated regulation of enzyme activity (revised Figure 6N). Additionally, based on the crystal structure (PDB ID: 1L1F), K545 is adjacent to the regulatory binding domain in GLUD1 (revised Supplementary Figure 6F). The regulatory domain is situated near the pivothelix between adjacent protomers. An activator could bind to the regulatory domain and hasten the opening of the catalytic cleft that leads to the release of the reaction product [6,7]. These data strongly suggest that K545 in GLUD1 is a major glutarylation target of SIRT5, and subsequently affects its activity. We have made extensive modifications in the results and discussion sections of the manuscript.

REFERENCES

6. Borgnia, M.J. et al. Using Cryo-EM to Map Small Ligands on Dynamic Metabolic Enzymes: Studies with Glutamate Dehydrogenase. *Mol Pharmacol* 89, 645-51 (2016).
7. Banerjee, S., Schmidt, T., Fang, J., Stanley, C.A. & Smith, T.J. Structural studies on

ADP activation of mammalian glutamate dehydrogenase and the evolution of regulation. *Biochemistry* 42, 3446-56 (2003).

5. Figure 7. The *in vivo* evidence would be more convincing if the authors determine the levels of TCA cycle metabolites in the xenograft models.

Author response:

According to your suggestion, we tested the levels of TCA cycle metabolites in the xenograft models. Nude mice (nu/nu, male, 5 weeks old) were injected subcutaneously with HCT116 cells stably expressing the non-target control (NTC) shRNA or SIRT5 shRNA (5×10^6 cells). As shown in revised Supplementary Figure 7A-D, the tumor volume and weight of the SIRT5 shRNA group were significantly decreased compared with that of the control shRNA group ($P < 0.05$). Consistent with our previous *in vitro* results, GC-MS analysis of tumor lysates also showed that knockdown of SIRT5 resulted in significantly downregulation of TCA cycle metabolites, including α -KG, succinate, fumarate, malate, citrate, and isocitrate (revised Supplementary Figure 7E& 7F).

Reviewer #2: (Remarks

to the Author):

In this study, Wang et al describe a role for SIRT5 in sustaining TCA pools via control of glutaminolysis via deglutarylation of GDH. Overall, the evidence that SIRT5 supports cell proliferation is strong and has been shown using multiple strategies. However, evidence supporting the underlying mechanism is less convincing. Additional studies directed at the direct effects of SIRT5 on metabolism and its influence over metabolic flux would significantly strengthen the work.

MAJOR:

1.-Cell proliferation. Given that cellular growth changes metabolic requirements, how much is growth rate playing a role in the isotope tracing effects observed? Faster growing cells (SIRT5 o/e) or slower growing cells (siSIRT5) would be

expected to behave exactly as shown here (more/less glutamine metabolism), independent of SIRT5. This is a key point, and is not discussed. This is especially relevant given that the effects are subtle. Could differences in growth rate be driving the differences in metabolic labeling?

Author response:

We agree with the reviewers' comments about the growth rate playing a role in the isotope tracing effects. As shown in Figure 2B, we found that SIRT5 depletion did not significantly suppress LoVo cell proliferation at 24 h after SIRT5 knockdown. To exclude metabolic changes caused by cell growth, we re-performed the glutamine isotope profiling analysis in LoVo cells at the time point when siRNAs suppressed SIRT5 expression, but not cell growth. Briefly, LoVo cells were transfected with *SIRT5* siRNAs and cultured overnight. The cell numbers at the indicated time points were counted after addition of ^{13}C labeled-glutamine (6, 18, and 24 h). The knockdown efficiency was validated by western blotting (Supplementary Figure 2A for reviewers). We confirmed that suppression of SIRT5 had no effect on cell growth at the indicated time points after incubation with $[\text{U-}^{13}\text{C}_5]\text{glutamine}$ (Supplementary Figure 2B for reviewers). In addition, knockdown of SIRT5 robustly decreased ^{13}C incorporation into the TCA cycle at either 18 h (Supplementary Figure 2C for reviewers) or 24 h (revised Figure 4D-G) after tracer medium treatment. These results confirmed that SIRT5 is responsible for the observed metabolic alterations, independent of its effect on cell growth. To avoid any bias in the analysis, the data were also normalized to the protein concentration.

Supplementary Figure 2 for reviewers

Supplementary Figure 2 for reviewers. (A) The knockdown efficiency of *SIRT5* siRNAs at the indicated time points after ¹³C labeled-glutamine treatment was validated by western blotting. (B) Cell numbers at the corresponding time after ¹³C labeled-glutamine addition were measured (n = 4). (C) The mass isotopolog distributions of glutamine-derived TCA cycle intermediates and aspartate were decreased in *SIRT5*-silenced cells compared with the control siRNA treated cells. Cells were cultured in the presence of 2 mM [U-¹³C₅]glutamine, and direct glutamine-derived metabolites were measured by GC-MS 18 h after the addition of ¹³C labeled-glutamine; n = 4, **P* < 0.05, ***P* < 0.01, ***, *P* < 0.001. Student's t-test. N.S. = not significant for the indicated comparison.

2.-Analyses of Isotope labeling. There are several problems with the way the isotope labeling data are presented and interpreted. First, it doesn't appear that the cells are in steady-state at the time points measured (Fig 4). The dramatic reduction in of isotope signal in the knock-down indicates this, even at the 3h-timepoint, which was subsequently used for over-expressor data. Further, the effect of SIRT5 on the dilution (wash-out) of isotope appears to be stronger than the labeling kinetics, which does not necessarily support their hypothesis. Ideally, the cells should be at a steady-state of growth to that fluxes can be interpreted. Additionally, changes in reductive carboxylation observed under SIRT5 overexpression are modest and do not seem to be enough to account for the significant changes in cell proliferation. Finally, all of the isotopologues should be shown (such as for the knock-down data), not just specific isotopes. At minimum, these should be included in the supplement to allow for interpretation of the data. Lastly, Fig. 4E doesn't appear to reflect 4D and 4F; where did this come from?

Author response:

Thank you for the critical but useful comments. First, to validate the cells were in steady-state, we measured mass isotopomer distributions (MIDs) of metabolites extracted from cells at later time points, e.g. 18 and 24 h after [U-¹³C₅]glutamine incubation. As shown in revised Supplementary Figure 2C, metabolite labeling did not change significantly between 18 and 24 h, indicating that the cells have achieved metabolic and isotopic steady state at the indicated time point. Therefore, we chose the time point of 24 h after the labeled glutamine incubation for subsequent experiments. Control or SIRT5 knockdown cells were cultured for 24 h in the presence of [U-¹³C₅]glutamine before metabolite extraction and GC-MS analysis. We observed that results that were in agreement with the original manuscript. SIRT5 knockdown reduced the direct glutamine contribution to α -KG (m+5) (revised Figure 4E). The fractions of succinate (m+4), fumarate (m+4), malate (m+4), citrate (m+4), isocitrate (m+4), and aspartate(m+4) were repressed in SIRT5 knockdown cells (revised Figure 4F &4G).

To analyze the overexpression data, steady-state labeling of TCA metabolites was accomplished by culturing cells stably expressing vector control, SIRT5 WT, and SIRT5 H158Y in tracer medium for 24 h. Similarly, we observed that SIRT5 overexpression increased the fraction of m+5 α -KG significantly (revised Figure 4I) and promoted a higher rate of incorporation of [U- $^{13}\text{C}_5$] glutamine into m+4-labeled TCA cycle intermediates, accompanied by glutamine-derived aspartate and asparagines (revised Figure 4J), while the enzymatically deficient mutant did not exhibit these function.

In our re-performed [U- $^{13}\text{C}_5$]glutamine labeling experiments, we confirmed the slightly reduced levels of citrate(m+5), fumarate (m+3), malate (m+3), and aspartate (m+3) upon SIRT5 knockdown. Given that citrate (m+5) can be formed through condensation of glutaminolysis-derived AcCoA (m+2) + oxaloacetate (m+3)[8], the reductive glutamine metabolism, especially in the presence of high glutaminolysis flux, is best assessed with [1- ^{13}C]glutamine, which transfers carbon to citrate only through the reductive carboxylation pathway [9] (revised Supplementary Figure 4A). Thus, we further tested m+1 label in metabolites pools derived from [1- ^{13}C]glutamine. As shown in revised Supplementary Figure 4B &4C, although significant alterations in m+1 α -KG levels were observed, neither knockdown nor overexpression of SIRT5 led to an observable difference in citrate (m+1), isocitrate (m+1), malate (m+1), fumarate (m+1), or aspartate (m+1) enrichment. Therefore, we corrected the related part about reductive carboxylation in the results section of the revised manuscript.

For the ^{13}C -based metabolic flux assay, we presented all of the isotopologs in the revised manuscript as the reviewer suggested (revised Figure 4D-H for the siRNA data and revised Supplementary Figure 3 for the overexpression data, respectively).

Figure. 4E displays the ratio of α -KG (m+5)/ glutamate (m+5) in SIRT5-silenced cells when normalized to control siRNA transfected cells. It was calculated using raw mass spectrometry data (isotopic clusters), instead of the mass isotopologue distribution. Considering that knockdown of SIRT5 did not affect ^{13}C -labeled glutamate while significantly reducing m+5 α -KG (revised Figure 4D and 4E), we speculated that SIRT5 might regulate the conversion of glutamate to α -KG, which was

verified in Figure 5. We have revised the manuscript and removed Fig 4E to make it clear.

REFERENCES

8. Montal, E.D. et al. PEPCCK Coordinates the Regulation of Central Carbon Metabolism to Promote Cancer Cell Growth. *Mol Cell* 60, 571-83 (2015).
9. Gameiro, P.A. et al. In vivo HIF-mediated reductive carboxylation is regulated by citrate levels and sensitizes VHL-deficient cells to glutamine deprivation. *Cell Metab* 17, 372-85 (2013).

3.-GLUD1 as a primary target. Chemical glutarylation of GLUD1 using glutaryl-CoA results in a slight reduction in its activity which cannot completely explain the significant lack of proliferation in absence of SIRT5. Blocking GDH with ECGC is sufficient to block all cell growth, independent of SIRT5, so while interesting, this reagent does not necessarily support a role for SIRT5 regulating GDH in this study. In the xenograft model increase in GLUD1 activity with SIRT5 overexpression is impressive but the authors do not show if there was an appreciable increase in GLUD1 expression in this model or is this increase in activity indeed due to SIRT5-dependent posttranslational modification. A detailed investigation of which sites on GLUD1 are important were not performed; identifying the sites of regulation would strengthen this argument.

Author response:

Thank you for the comments. We found a direct interaction between SIRT5 and GLUD1, which causes the deglutarylation of GLUD1. Given that GLUD1 is activated by SIRT5 in a deacylation-dependent manner, we set out to examine whether lysine glutarylation or other acylation modifications, such as succinylation or malonylation, would alter its activation. Our results demonstrated that the activity of GLUD1 was significantly decreased after incubation with glutaryl-CoA *in vitro* but was unaffected by succinyl-CoA or malonyl-CoA, indicating that glutarylation of GLUD1 may negatively regulate its activation. The immunoprecipitated HA-tagged GLUD1 was

incubated with acyl-CoA *in vitro*, followed by determination of its enzyme activity. Considering that chemical acylation of the target proteins is sensitively dependent on physiological pH, the period of incubation and acyl-CoA concentrations [10,11], in the re-performed assay, we optimized the conditions by decreasing the ratios of protein/acyl -CoA and prolonged the incubation time. As shown in revised Figure 6L, glutaryl-CoA incubation in HCT116 and LoVo cells decreased the activity of GLUD1 by 20% and 25%, respectively (10% and 11% in the original manuscript); whereas, no significant difference was observed in the succinyl-CoA or malonyl-CoA treatment groups under the same conditions (revised Supplementary Figure 6D). Similarly, a recent study revealed that IDH could be succinylated and succinyl-CoA incubation resulted in at most 20% reduction of its activity [12].

We agree with the reviewer's suggestion. EGCG is a highly efficient but not specific inhibitor of GLUD. As pointed out by the reviewer, EGCG itself is sufficient to block all cell growth. Thus, we no longer include this panel in the revised Figure 5H.

According to the reviewer's advice, we detected GLUD1 protein levels in tumor lysates derived from xenografts. As shown in revised Figure 7F, we did not observe a significant change of GLUD1 expression in SIRT5 WT or SIRT5 H158Y plasmid treated tumors. Collectively, we confirmed that SIRT5 increased GLUD1 activity rather than upregulated its protein level *in vivo*.

A similar question about the sites of regulation on GLUD1 was raised by reviewer #1. We have further determined the SIRT5-dependent deglutarylation sites in GLUD1 and have made extensive modifications in the results and discussion sections. As reported in a previous proteomic study (Supplementary Table 3), there are nine different lysine residues in GLUD1 that could potentially be modified by glutarylation. Among them, lysine 399, lysine 503, and lysine 545 are conserved in GLUD1 orthologs from human to *Drosophila melanogaster*, indicating that these residues may be critical to some evolutionarily conserved functions of GLUD1 (revised Supplementary Figure 6E). The lysine (K) to arginine (R) mutation retains a positive charge and is often utilized as a deacylated mimetic. Therefore, we generated three

plasmids encoding mutant HA-tagged GLUD1, in which lysine 399, lysine 503, or lysine 545 residues were substituted by arginines (R), respectively. Ectopically expressed wild-type GLUD1, and the K399R, K503R, and K545R mutants were transfected into HCT116 cells, followed by SIRT5 knockdown. The glutarylations of GLUD1 were analyzed by western blotting. We found that the K545R mutation resulted in a significant reduction of glutarylation. Notably, SIRT5 suppression increased the glutarylation levels of wild-type GLUD1, and the K399R and K503R mutant, but not the K545R mutant, indicating that GLUD1 was glutarylated in a SIRT5-dependent manner on lysine 545 (revised Figure 6M). Consistent with this, K545R mutant GLUD1 displayed no response to SIRT5-mediated regulation of enzyme activity (revised Figure 6N). Additionally, based on the crystal structure (PDB ID: 1L1F), K545 is adjacent to the regulatory binding domain in GLUD1 (revised Supplementary Figure 6F). The regulatory domain is situated near the pivothelix between adjacent protomers. An activator could bind to the regulatory domain and hasten the opening of the catalytic cleft that leads to the release of the reaction product [6,7]. These data strongly suggest that K545 in GLUD1 is a major glutarylation target of SIRT5, and subsequently affect its activity.

REFERENCES

10. Wagner, G.R. & Payne, R.M. Widespread and enzyme-independent Nepsilon-acetylation and Nepsilon-succinylation of proteins in the chemical conditions of the mitochondrial matrix. *J Biol Chem* 288, 29036-45 (2013).
11. Paik, W.K., Pearson, D., Lee, H.W. & Kim, S. Nonenzymatic acetylation of histones with acetyl-CoA. *Biochimica Et Biophysica Acta* 213, 513 (1970).
12. Zhou, L. et al. SIRT5 promotes IDH2 desuccinylation and G6PD deglutarylation to enhance cellular antioxidant defense. *EMBO Rep* 17, 811-22 (2016).
6. Borgnia, M.J. et al. Using Cryo-EM to Map Small Ligands on Dynamic Metabolic Enzymes: Studies with Glutamate Dehydrogenase. *Mol Pharmacol* 89, 645-51 (2016).
7. Banerjee, S., Schmidt, T., Fang, J., Stanley, C.A. & Smith, T.J. Structural studies on ADP activation of mammalian glutamate dehydrogenase and the evolution of

regulation. *Biochemistry* 42, 3446-56 (2003).

MINOR:

4.-The paper is set-up with the concept and findings that SIRT5 is over-expressed and over-abundant in CRC, and correlates with poor prognosis. But then the authors move onto studies with SIRT5 silencing, and then make the conclusion that “inhibition of apoptosis and promotion of cell cycle progression could explain the observed SIRT5-induced proliferation of CRC cells”. However, this is never directly tested. If possible, the authors should directly test their conclusions, rather than infer them from the opposing result (e.g. if knock-down leads to less growth and more apoptosis, then higher expression explains more growth and less apoptosis). The authors began to test some phenotypes directly in Fig. 3, but never tested cell cycle or apoptosis in their over-expressing lines. The inconsistencies in measurements from the over-expressers and knock-downs is puzzling.

Author response:

We would like to thank the reviewer for these insightful comments, which greatly helped to improve our manuscript. We corrected the sentence "The inhibition of apoptosis and promotion of cell cycle progression could explain the observed SIRT5-induced proliferation of CRC cells" to " Thus, SIRT5 silencing inhibited CRC cell proliferation by inducing apoptosis and cell cycle arrest."

Consistent with our RNAi knockdown data, we found that forced expression of SIRT5 substantially inhibited apoptosis of HCT116 and LoVo cells, along with a downregulation of cleaved caspase 3, caspase 8 (active fragment p18), and PARP (Supplementary Figure 3A-C for reviewers). Furthermore, the growth curves of HCT116 and LoVo cells revealed that SIRT5 knockdown significantly suppressed cell proliferation, whereas its overexpression markedly promoted proliferation (Supplementary Figure 3D& 3E for reviewers). All these results are in line with our findings that overexpression and over-abundance of SIRT5 promote CRC cells proliferation, thus contributing to the malignant phenotype of CRC.

Supplementary Figure 3 for reviewers

Supplementary Figure 3 for reviewers. (A) Flow cytometric assay based on Phycoerythrin-conjugated annexin V staining showing that SIRT5 WT overexpression attenuated the serum deprivation-induced apoptosis in HCT116 and LoVo cells compared with the vector and SIRT5 H158Y clones. Representative fluorescence activated cell sorting (FACS) images are shown. (B) Quantified results of apoptosis analysis in HCT116 and LoVo cells stably expressing the control vector, SIRT5 WT,

and SIRT5 H158Y plasmid. Results are presented as the mean \pm SD of three independent samples. *P*-values were calculated by ANOVA with Tukey's test. **, *P* < 0.05 and ***, *P* < 0.001. N.S. = not significant for the indicated comparison. (C) Western blotting analysis of HCT116 and LoVo cells stably expressing the control vector, SIRT5 WT, and SIRT5 H158Y confirmed the altered levels of cleaved caspase 3, caspase 8, and PARP. (D-E) Proliferation curves of HCT116 (D, left) and LoVo (E, left) cells after transfection of control siRNA or *SIRT5* siRNAs. Proliferation curves of HCT116 (D, right) and LoVo (E, right) cells stably expressing control vector or SIRT5 WT plasmid. Student's t-test. ***, *P* < 0.001.

5.-During the initial screen of the CRC tissue authors point out that the expression of SIRT5 is mainly cytoplasmic however later they attribute the mechanism of increased proliferation of CRC cell lines to changes in a post-translational modification, protein glutarylation, which is primarily mitochondrial on a protein GLUD1 which again is primarily mitochondrial. Although authors later show co-localization of GLUD1 and SIRT5 in the CRC cell lines how do the authors reconcile the difference in expression of SIRT5 in cells Vs CRC tissue?

Author response:

Thank you for these helpful comments. SIRT5 mainly localizes in the mitochondrial matrix, with a small portion existing extra-mitochondrially [13,14]. To localize endogenous SIRT5 *in vivo*, we performed immunofluorescence and confocal microscopy in CRC tissues. As shown in revised Figure 1C, we identified that the green fluorescence caused by SIRT5 was superimposed with the red fluorescence caused by Mito-track (Anti-Mitochondria antibody), suggesting that the majority of SIRT5 is located in the mitochondria. Additionally, in agreement with the observed co-localization in CRC cell lines, we also confirmed the strong spatial overlap between GLUD1 and SIRT5 in CRC tissues (revised Supplementary Figure 6B). Therefore, these results confirmed a direct interaction between SIRT5 and GLUD1 *in vivo*.

REFERENCES

13. Nakagawa, T., Lomb, D.J., Haigis, M.C. & Guarente, L. SIRT5 Deacetylates carbamoyl phosphate synthetase 1 and regulates the urea cycle. *Cell* 137, 560-70 (2009).
14. Park, J. et al. SIRT5-mediated lysine desuccinylation impacts diverse metabolic pathways. *Mol Cell* 50, 919-30 (2013).

Reviewer #3:(Remarks to the Author):

The manuscript by Wang et al. describes the role of Sirtuin5 in malignant colorectal cancer through its activation of GLUD1, which converts glutamate to α -ketoglutarate. The authors begin by showing increased SIRT5 expression in human tissues through their own data and publicly available GEO datasets. To confirm biological relevance of SIRT5, the authors performed siRNA knock-down (KD) of SIRT5 in 2 cell lines and showed decreased proliferation, cell cycle arrest, and increased apoptosis in the KD cells. The converse experiment with overexpression of wild-type SIRT5 or catalytically inactive SIRT5 showed a growth improvement with wild-type SIRT5 overexpression. The authors also traced the metabolism of [U-¹³C₅]glutamine in SIRT5 KD cells and showed a reduction in TCA cycle labeling from citrate independent of changes in glutamate labeling. The authors also worked to identify the mechanism of GLUD1 deglutarylation through direct interaction with SIRT5. Finally, the authors show robust tumor growth of SIRT5 wild-type overexpressing tumors compared to catalytically inactive SIRT5 tumors. Furthermore, this growth advantage of wild-type overexpressing SIRT5 is ablated with shRNA KD of GLUD1. This manuscript is well organized and clearly explains the experimental designs and data interpretation. Some revisions are needed, however, before acceptance for publication.

Major points:

1. • To what extent does glutaminolysis extend to pyruvate and lactate labeling? Do the authors see pyruvate or lactate labeling from [U-¹³C₅]glutamine?

Author response:

We thank the reviewer for this valuable suggestion. In transformed tumor cells, glutamine can be partially oxidized to pyruvate and lactate via flux through the TCA cycle (revised Figure 4B) [15]. We further tested glutamine-derived pyruvate and lactate in the revised manuscript. As shown in revised Figure 4H, although a small percent of pyruvate and lactate was derived from glutamine (< 5%), suppression of SIRT5 led to a slight but significant decrease in pyruvate (m+3) and lactate (m+3) (P < 0.01). In contrast, overexpression of SIRT5 WT, but not SIRT5 H158Y, increased the pyruvate and lactate labeling from [U-¹³C₅] glutamine (revised Supplementary Figure 3C). The increase in lactate and pyruvate derived from glutamine further confirmed that SIRT5 could enhance the TCA cycle flux.

REFERENCES

15. De Berardinis, R.J. et al. Beyond aerobic glycolysis: transformed cells can engage in glutamine metabolism that exceeds the requirement for protein and nucleotide synthesis. *Proc Natl Acad Sci U S A* 104, 19345-50 (2007).

2. • Following this point, M+5 citrate and m+3 malate, fumarate, and aspartate are used to indicate reductive glutamine metabolism. However, m+5 citrate can be formed through condensation of glutaminolysis-derived pyruvate AcCoA + m+3 oxaloacetate. Reductive glutamine metabolism to citrate, especially in the presence of high anaplerotic glutamine flux or high malic enzyme activity, is best assessed with [1-¹³C]glutamine.

Author response:

We appreciate the reviewer's helpful comments. In our re-performed [U-¹³C₅]glutamine labeling experiments, we confirmed the slightly reduced levels of citrate (m+5), fumarate (m+3), malate (m+3), and aspartate (m+3) upon SIRT5 knockdown. As pointed out by the reviewer, citrate (m+5) would result from

glutaminolysis-derived AcCoA (m+2) condensing with oxaloacetate (m+3) [8]. To accurately monitor reductive glutamine metabolism, we conducted a metabolic flux assay with [1-¹³C]glutamine, which transfers carbon to citrate only through the reductive carboxylation pathway [9] (revised Supplementary Figure 4A). The flow of the isotopic glutamine tracer is shown in revised Supplementary Figure 4B & 4C. Although significant alterations in m+1 α -ketoglutarate levels were observed, neither knockdown nor overexpression of SIRT5 changed the levels of labeled m+1 malate, fumarate, and aspartate derived from [1-¹³C]glutamine. Therefore, our results confirmed that SIRT5 mainly regulated oxidative glutamine metabolism in CRC cells, but had no obvious effect on reductive carboxylation. We have modified the related parts of the revised manuscript.

REFERENCES

8. Montal, E.D. et al. PEPCK Coordinates the Regulation of Central Carbon Metabolism to Promote Cancer Cell Growth. *Mol Cell* 60, 571-83 (2015).
9. Gameiro, P.A. et al. In vivo HIF-mediated reductive carboxylation is regulated by citrate levels and sensitizes VHL-deficient cells to glutamine deprivation. *Cell Metab* 17, 372-85 (2013).

3. The authors see robust labeling of TCA cycle intermediates on these short time scales. It may present a more complete picture of total tracer incorporation to plot each intermediate's mole percent enrichment from [U-¹³C₅]glutamine.

Author response:

Thank you for the advice. In the revised version, we have prolonged the observation time and validated that the cells were in steady state. Cells were cultured for 24 h in the labeled glutamine before metabolite extraction and GC-MS analysis. The results (revised Figure 4) are consistent with those of original manuscript. Furthermore, as the reviewer suggested, the relative abundance of different mass isotopologues of each metabolite were presented (revised Figure 4D-H for the siRNA data and revised Supplementary Figure 3 for the overexpression data, respectively).

4. • Do the authors have a theory for why the conversion of glutamate to α -KG is required for the pro-survival effect of SIRT5 and not simply maintenance of the α -KG pool? Is glutamate limiting in SIRT5 overexpression cells (similar to how SIRT5 KD causes an increase in glutamate levels)?

Author response:

We thank the reviewer for these critical comments. As shown in figure 5G, we observed that addition of glutamine downstream metabolites, including glutamate, dimethyl- α -KG, and NEAA, promoted the growth of CRC cells in the glutamine-depleted media. However, only glutamate (the upstream substrate of the conversion), but not dimethyl- α -KG or NEAA (the downstream product of the conversion) restored the SIRT5-dependent cancer cell proliferation. By contrast, both GLUD1 and aminotransaminases are responsible for the conversion of glutamate to α -KG, and contribute to maintenance of the α -KG pool. However, we demonstrated that GLUD1, but not aminotransaminases, is critical and sufficient for SIRT5-mediated cancer progression (revised Figure 5H and Supplementary Figure 5D). Based on these results, we proposed that the conversion of glutamate-to- α -KG conducted by GLUD1 is required for the pro-survival effect of SIRT5. Furthermore, the conversion catalyzed by GLUD1 accompanied the course of maintaining NADPH production [16,17]. NADPH is required for redox control and cancer cell survival [18]. Additionally, a recent study reported that knockout of SIRT5 resulted in decreased NADPH production and impaired proliferation, which also supports our hypothesis [12]. As the reviewer suggested, we have more carefully discussed and added the detailed information to paragraph 4 of the discussion section.

Thank you for the suggestion. We assayed glutamate levels using a Glutamate Assay Kit (catalog #EGLN-100, Bioassay systems, Hayward, CA, USA). As shown in Supplementary Figure 4 for reviewers, after 12 h of incubation in fresh media, we found increased levels of extracellular glutamate in CRC cell supernatants upon SIRT5 KD. Conversely, overexpression of SIRT5 WT decreased the amount of

glutamate secreted into the culture media compared with the control vector and SIRT5 H158Y transfected cells.

REFERENCES

16. Lorin, S. et al. Glutamate dehydrogenase contributes to leucine sensing in the regulation of autophagy. *Autophagy* 9, 850-860 (2014).
17. Villar, V.H., Merhi, F., Djavaheiri-Mergny, M. & Duran, R.V. Glutaminolysis and autophagy in cancer. *Autophagy* 11, 1198-208 (2015).
18. Jeon, S.-M., Chandel, N.S. & Hay, N. AMPK regulates NADPH homeostasis to promote tumour cell survival during energy stress. *Nature* 485, 661-665 (2012).
12. Zhou, L. et al. SIRT5 promotes IDH2 desuccinylation and G6PD deglutarylation to enhance cellular antioxidant defense. *EMBO Rep* 17, 811-22 (2016).

Supplementary Figure 4 for reviewers

Supplementary Figure 4 for reviewers. HCT116 and LoVo cells treated with *SIRT5* siRNA (A) or stably expressing the control vector, *SIRT5* WT and *SIRT5* H158Y (B) were incubated with fresh media for 12 h, and the amount of glutamate in the culture media was assayed using a Glutamate Assay Kit. Results were normalized to the control groups; n = 4. ANOVA with Tukey's test. *, $P < 0.05$, ** $P < 0.01$, *** $P < 0.01$.

Minor points:

5. In plots of cell growth the y-axis is labeled as "proliferation rate." Is this raw cell

number or an actual proliferation rate? If this is a calculated rate an equation should be included in the methods.

Author response:

We apologize for this mistake. We measured the cell numbers spectrophotometrically using a cell counting kit-8. Thus, the y-axis should be labeled as OD 450 absorbance. According to your advice, we have amended the relevant parts in the revised manuscript. Only the results described in Figure 5G were calculated as the proliferation rate. In addition, we modified the related parts of methods as follows: Cells were seeded onto 96 wells culture plates overnight. The next day, CCK-8 solution was added and the absorbance at 450 nm was determined (OD 450 absorbance DAY 0). Media was changed according to different culture conditions following 4 days of incubation. At the end of the study, the absorbance at 450 nm was measured (OD 450 absorbance DAY 4). The proliferation rate = $\text{OD 450 absorbance DAY 4} - \text{blank} / \text{OD 450 absorbance DAY 0} - \text{blank}$.

6. • Potential error in Supplementary Fig 2C: y-axis label of %m+5 does not seem possible given the time of cells in culture and compared to other Fig 4 graphs.

Author response:

We are sorry for this confusion. We have checked and confirmed the data concerning the fraction of m+5 glutamine in revised Supplementary Fig 3D (Supplementary Fig 2C in the original manuscript). The fraction of m+5 glutamine reflected the glutamine directly taken up by cancer cells, which is the first step of glutaminolysis. The added [U-¹³C₅] glutamine (2 mM) was in excess and was sufficient for CRC cell proliferation, and could be replenished immediately in the cells. Other graphs in Figure 4 reflected the metabolites in the subsequent metabolic process of glutaminolysis.

7. • Fig 5G should reflect that NEAAs added were only aspartate and asparagine.

Author response:

We have modified the legends in Figure 5G according to reviewer's suggestion (see revised Figure 5G).

8. • In figures with many t-tests, especially those where multiple bars are compared to a single control condition, the authors should use an ANOVA or at least correct for multiple comparisons within a single figure panel.

Author response:

Thanks you for the valuable suggestion. We have checked throughout the manuscript and corrected the statistical methods for all the experiments. Comparisons of only two conditions were performed using Student's t test (two-tailed). For multiple comparisons, an ANOVA was used for statistical analysis. In addition, we have amended the relevant parts in the revised manuscript.

REVIEWERS' COMMENTS:

Reviewer #1 (Remarks to the Author):

The authors has appropriately addressed the comments raised by the reviewers.

Reviewer #2 (Remarks to the Author):

The authors adequately addressed my previous concerns.

Reviewer #3 (Remarks to the Author):

The authors have addressed my concerns.

Response to reviewers

Reviewer #1 (Remarks to the Author):

The authors has appropriately addressed the comments raised by the reviewers.

Our response: Thanks very much. We are delighted that we could address issues clearly.

Reviewer #2 (Remarks to the Author):

The authors adequately addressed my previous concerns.

Our response: Thanks very much for your comments. Once more we appreciate the time and effort made by this reviewer

Reviewer #3 (Remarks to the Author):

The authors have addressed my concerns.

Our response: We thanks very much for your second review of the manuscript.